# Phase transitions for the existence of unregularized M-estimators in single index models

**Takuya Koriyama** [1]   **Pierre C. Bellec** [2]

## Abstract

This paper studies phase transitions for the existence of unregularized M-estimators under proportional asymptotics where the sample size $n$ and feature dimension $p$ grow proportionally with $n/p \to \delta \in (1, \infty)$. We study the existence of M-estimators in single-index models where the response $y_i$ depends on covariates $\boldsymbol{x}_i \sim N(0, I_p)$ through an unknown index $\boldsymbol{w} \in \mathbb{R}^p$ and an unknown link function. An explicit expression is derived for the critical threshold $\delta_\infty$ that determines the phase transition for the existence of the M-estimator, generalizing the results of Candès & Sur (2020) for binary logistic regression to other single-index models. Furthermore, we investigate the existence of a solution to the nonlinear system of equations governing the asymptotic behavior of the M-estimator when it exists. The existence of solution to this system for $\delta > \delta_\infty$ remains largely unproven outside the global null in binary logistic regression. We address this gap with a proof that the system admits a solution if and only if $\delta > \delta_\infty$, providing a comprehensive theoretical foundation for proportional asymptotic results that require as a prerequisite the existence of a solution to the system.

## 1. Introduction

Let $(\boldsymbol{x}_i, y_i)_{i \in [n]}$ be a sample of i.i.d. observations where $\boldsymbol{x}_i \in \mathbb{R}^p$ and follows the normal distribution $\boldsymbol{x}_i \sim N(0, I_p)$. We consider responses $y_i \in \mathcal{Y}$ where $\mathcal{Y} \subset \mathbb{R}$ and assume a single-index model of the form

$$\mathbb{P}(y_i \le t \mid \boldsymbol{x}_i) = F(t, \boldsymbol{x}_i^\top \boldsymbol{w}), \qquad \forall t \in \mathbb{R}$$

[1]University of Chicago, Booth School of Business [2]Department of Statistics, Rutgers University. Correspondence to: Takuya Koriyama <tkoriyam@uchicago.edu>.

*Proceedings of the 42$^{nd}$ International Conference on Machine Learning*, Vancouver, Canada. PMLR 267, 2025. Copyright 2025 by the author(s).

where $F : \mathbb{R} \times \mathbb{R} \to [0, 1]$ is an unknown deterministic function and $\boldsymbol{w} \in \mathbb{R}^p$ is an unknown index with $\|\boldsymbol{w}\| = 1$. This includes generalized linear models, such as the Poisson model (in which $\mathcal{Y} = \{0, 1, 2, \dots\}$) or Binary logistic model (in which $\mathcal{Y} = \{-1, 1\}$ for instance).

An unregularized M-estimator is fit to observed data $(y_i, \boldsymbol{x}_i)_{i \in [n]}$ by the minimization problem

$$\inf_{\boldsymbol{b} \in \mathbb{R}^p} \sum_{i=1}^n \ell_{y_i}(\boldsymbol{x}_i^\top \boldsymbol{b}) \tag{1}$$

where $\ell_y(\cdot)$ is convex for every $y \in \mathcal{Y}$. If the infimum is achieved, we say the M-estimator exists and denote it by $\hat{\boldsymbol{b}}$, i.e.,

$$\hat{\boldsymbol{b}} \in \mathrm{argmin}_{\boldsymbol{b} \in \mathbb{R}^p} \sum_{i=1}^n \ell_{y_i}(\boldsymbol{x}_i^\top \boldsymbol{b}).$$

In this paper we focus on a high dimensional regime where the sample size and feature grow proportionally as

$$\frac{n}{p} \to \delta \in (1, +\infty)$$

Here the constant $\delta$ quantifies sample size per dimensions. In this proportional asymptotic regime, under logistic regression model with binary response $\mathcal{Y} = \{-1, 1\}$ and with $\ell_y(t) = \log(1 + \exp(-yt))$ the logistic loss, the seminal work of Candès & Sur (2020) establishes that the existence of the M-estimator undergoes a sharp phase transition at a critical threshold $\delta_\infty$:

- if $\delta > \delta_\infty$ then the M-estimator exists with high-probability (i.e., the infimum in (1) is attained), while

- if $\delta < \delta_\infty$ then the M-estimator does not exist (i.e., the infimum is not attained) with high-probability.

If $\delta > \delta_\infty$, the behavior of the unregularized M-estimator $\hat{\boldsymbol{b}}$, including the limit in probability of $\hat{\boldsymbol{b}}^\top \boldsymbol{w}$ and the limit in probability of $\|(I_p - \boldsymbol{w}\boldsymbol{w}^\top)\hat{\boldsymbol{b}}\|_2$, is characterized (Sur & Candès, 2019) by the nonlinear system with three unknowns $(\gamma, a, \sigma) \in \mathbb{R}_{>0} \times \mathbb{R} \times \mathbb{R}_{>0}$:

$$\gamma^{-2}\delta^{-1}\sigma^2 = \mathbb{E}[\ell'_Y(\mathrm{prox}[\gamma\ell_Y](aU + \sigma G))^2],$$
$$0 = \mathbb{E}[U\ell'_Y(\mathrm{prox}[\gamma\ell_Y](aU + \sigma G))], \tag{2}$$
$$\sigma(1 - \delta^{-1}) = \mathbb{E}[G\,\mathrm{prox}[\gamma\ell_Y](aU + \sigma G)],$$

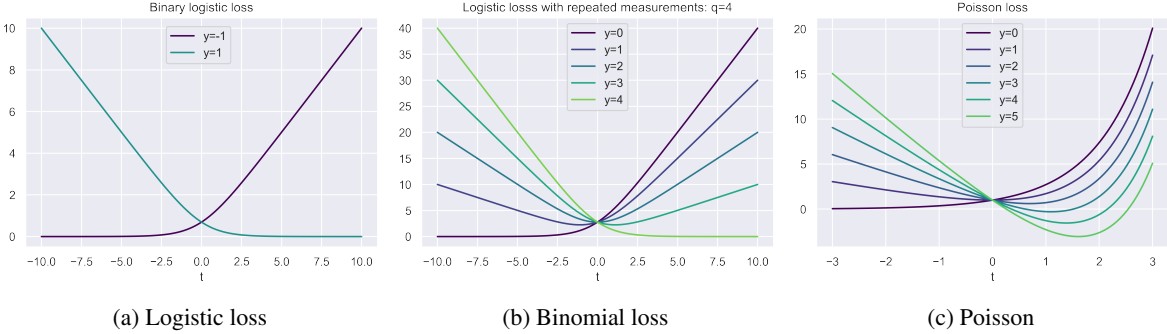

(a) Logistic loss          (b) Binomial loss          (c) Poisson

*Figure 1.* Three examples of loss functions.

where $G \sim N(0, 1)$ is independent of $(U, Y)$, and $(U, Y)$ has the same distribution as $(\boldsymbol{x}_i^\top \boldsymbol{w}, y_i)$. In particular $U \sim N(0, 1)$. The phase transition result of Candès & Sur (2020) for Gaussian design and binary logistic regression has been extended by Tang & Ye (2020) to elliptic covariate distributions and general binary response models. Han & Ren (2022) extended the phase transition of the logistic regression model with Gaussian covariates to the constrained minimization setting where the infimum in (1) is restricted to a closed convex cone. More broadly, the phase transition phenomena studied in our paper are connected to earlier works in statistical physics ((Cover, 1965; Gardner & Derrida, 1988; Krauth & Mézard, 1989)). Especially, (Cover, 1965) analyzed the geometry of linear inequalities and derived $\delta_\infty = 2$ under the null, i.e., when $\boldsymbol{x}_i$ is independent of $y_i$. More recently, similar phase transition behavior has been investigated in (Mignacco et al., 2020) for Gaussian mixtures models and in (Gerace et al., 2020) for random feature models.

Results such as Sur & Candès (2019); Salehi et al. (2019) studying the behavior of the M-estimator on the side of the phase transition where it exists with high-probability assume that the nonlinear system (2) admits a unique solution. Under the global null in binary logistic regression, Sur et al. (2019) establishes that $\delta_\infty = 2$ and that the system (2) admits a unique solution if and only if $\delta > 2$. Beyond the global null, it was observed (Sur & Candès, 2019) that the system (2) can be solved numerically if $\delta > \delta_\infty$ (where $\delta_\infty$ is characterized in Candès & Sur (2020)), and that if the solution exists, it is unique (see Remark 2 of the supplement of Sur & Candès (2019)). However to our knowledge there is no proof yet that the system admits a solution for $\delta > \delta_\infty$ except under the global null (see discussion after eq. (16) of the supplement of Sur & Candès (2019)).

The goal of the present paper is twofold:

- To characterize the critical threshold $\delta_\infty$ for Gaussian covariates beyond binary response models, for instance the Poisson model.

- To prove that the system (2) admits a unique solution if and only if $\delta > \delta_\infty$.

## 2. Main Result

Let us introduce the three examples of interest that our assumptions will cover.

*Example* 2.1 (Binary logistic regression). Here, labels in $\mathcal{Y} = \{-1, 1\}$ and the loss function

$$\ell_y(t) = \log(1 + \exp(-yt)).$$

*Example* 2.2 (Logistic regression with repeated measurements). Let $q \geq 2$. Here, $\mathcal{Y} = \{0, 1, \ldots, q\}$ and the loss function, corresponding to a binomial regression model with $q$ throws and a sigmoid link function for the probability, is

$$\ell_y(t) = q \log(1 + \exp(t)) - yt.$$

If $q = 1$, this is equivalent to binary logistic regression by renaming $\{1, 0\}$ to $\{1, -1\}$.

*Example* 2.3 (Poisson regression). If the labels are in $\mathcal{Y} = \{0, 1, 2, 3, \ldots\} = \mathbb{N}$, the non-negative likelihood of Poisson generalized linear model leads to the loss function

$$\ell_y(t) = \exp(t) - yt.$$

The phenomenon of the phase transition for the existence of the M-estimator comes from the lack of coercivity of some of the loss functions $\ell_{y_i}$ appearing in the optimization problem (1). For instance,

- In the binary logistic regression case (Example 2.1), the loss $\ell_{y_i}$ is not coercive for all $y_i$: it is increasing for $y_i = -1$ and decreasing for $y_i = 1$.

- For binomial logistic regression with $q \geq 2$ measurements, (Example 2.2), the loss $\ell_{y_i}$ is coercive if $y_i \in \{1, \ldots, q - 1\}$, increasing for $y_i = 0$ and decreasing for $y_i = q$.

- For Poisson regression, (Example 2.3), the loss $\ell_{y_i}$ is coercive for $y_i \geq 1$ and increasing for $y_i = 0$.

The values of $y_i$ leading to a coercive, increasing or decreasing loss $\ell_{y_i}(\cdot)$ and the distribution of $(\boldsymbol{x}_i^\top \boldsymbol{w}, y_i)$ will determine the critical threshold $\delta_\infty$. In order to study $\delta_\infty$, it will be thus useful to introduce the following notation: For a random variable $Y$ valued in $\mathcal{Y}$, we define the events $\Omega_\vee(Y), \Omega_\nearrow(Y), \Omega_\searrow(Y)$ by

$$
\begin{aligned}
\Omega_\vee(Y) &= \{\ell_Y(\cdot) \text{ is coercive}\}, \\
\Omega_\nearrow(Y) &= \{\ell_Y \text{ is strictly increasing}\}, \\
\Omega_\searrow(Y) &= \{\ell_Y \text{ is strictly decreasing}\}.
\end{aligned}
\tag{3}
$$

These events can be made explicit for the three examples thanks to the discussion in the three bullet points above:

- In the binary logistic regression case (Example 2.1) we have $\Omega_\nearrow(Y) = \{Y = -1\}$ and $\Omega_\searrow(Y) = \{Y = 1\}$.

- In Example 2.2, $\Omega_\nearrow(Y) = \{Y = 0\}$ as well as $\Omega_\searrow(Y) = \{Y = q\}$ and $\Omega_\vee(Y) = \{0 < Y < q\}$.

- For Poisson regression (Example 2.3) we have $\Omega_\nearrow(Y) = \{Y = 0\}$ and $\Omega_\vee(Y) = \{Y \geq 1\}$.

We will assume that $\ell_Y$ is strictly convex in our working assumptions, so that if $\ell_Y$ is not coercive it must be either strictly increasing or strictly decreasing. We first state an assumption that prevents the problem from being trivial.

**Assumption 2.4.** The loss $\ell_y$ is strictly convex for every $y \in \mathcal{Y}$, and the law of $Y$ satisfies

$$
\mathbb{E}\Big[U^2\big(I\{\Omega_\vee\} + I\{\Omega_\searrow, U > 0\} + I\{\Omega_\nearrow, U < 0\}\big)\Big] > 0,
$$
$$
\mathbb{E}\Big[U^2\big(I\{\Omega_\vee\} + I\{\Omega_\searrow, U < 0\} + I\{\Omega_\nearrow, U > 0\}\big)\Big] > 0
$$

where we omit the argument $Y$ in the three events $\Omega_\vee(Y), \Omega_\nearrow(Y)$ and $\Omega_\searrow(Y)$ for brevity.

This prevents the problem from being trivial in the following sense. Consider $p = 1$, and write $x_i = U_i$ (which is now scalar valued). Assume that the first line in Assumption 2.4 is 0. The minimization problem (5) becomes

$$
\inf_{a \in \mathbb{R}} \sum_{i=1}^n \ell_{y_i}(aU_i).
$$

Since $\mathbb{P}(U^2 > 0) = 1$ by $U \sim N(0, 1)$, this implies $\mathbb{P}(\Omega_\vee(Y)) = 0$, so for each of the $n$ terms, the loss $\ell_{y_i}(\cdot)$ is not coercive. In fact, each term is increasing in $a$, because $U_i > 0 \Rightarrow \Omega_\nearrow(y_i)$ and $U_i < 0 \Rightarrow \Omega_\searrow(y_i)$ by the first line in Assumption 2.4, so in this case $a \mapsto \sum_{i=1}^n \ell_{y_i}(aU_i)$ is increasing with probability one and the infimum is never

attained. Similarly, if the second line in Assumption 2.4 is zero, then $a \mapsto \sum_{i=1}^n \ell_{y_i}(aU_i)$ is decreasing and the infimum is never attained.

In conclusion, Assumption 2.4 merely assumes that for $p = 1$, the infimum is achieved, i.e., the M-estimator exists, with positive probability. If the M-estimator does not exist for $p = 1$ then it will not exist for $p > 1$ either, so Assumption 2.4 is required to avoid this trivial case that the M-estimator does not exist for all $p \geq 1$. The remaining of our working assumptions are given below.

**Assumption 2.5.** The loss $\ell_Y$ satisfies the following:

1. For all $y \in \mathcal{Y}$, $\ell_y : \mathbb{R} \to \mathbb{R}$ is $C^1$, strictly convex, and not constant.

2. $\mathbb{E}[|\inf_u \ell_Y(u)|] < +\infty$ and $\mathbb{E}[|\ell_Y(G)|] < +\infty$ where $G \sim N(0, 1)$ independent of $Y$.

3. $\mathbb{P}(\Omega_\vee) < 1$.

4. There exists a positive constant $b$ and $\sigma(Y)$-measurable positive random variable $D(Y)$ satisfying $\mathbb{E}[D(Y)] < +\infty$ and $\mathbb{E}[D^2(Y)\Omega_\vee(Y)] < +\infty$ such that

$$
\forall u \in \mathbb{R}, \quad \ell_Y(u) \geq -D(Y) + \frac{1}{b} \times
\begin{cases}
u & \text{under } \Omega_\nearrow \\
|u| & \text{under } \Omega_\vee \\
-u & \text{under } \Omega_\searrow
\end{cases}
$$

Beyond Assumption 2.4, Assumption 2.5 requires differentiability of the loss (item 1), mild integrability conditions (item 2), that $\ell_{y_i}$ is not always coercive (item 3) and that in the directions where $\ell_Y$ diverges to $+\infty$, it does so at least as fast as an affine function with slope $1/b$ and squared integrable intercept $D(Y)$ (item 4).

We define the critical threshold $\delta_\infty \in (0, +\infty]$ by

$$
\frac{1}{\delta_\infty} := \inf_{t \in \mathbb{R}} \varphi(t),
\tag{4}
$$

where $\varphi : \mathbb{R} \to \mathbb{R}_{\geq 0}$ is the convex function defined as

$$
\begin{aligned}
\varphi(t) := \ & \mathbb{E}\Big[\big(G + Ut\big)^2 I\{\Omega_\vee\}\Big] \\
& + \mathbb{E}\Big[\big(G + Ut\big)_+^2 I\{\Omega_\nearrow\}\Big] \\
& + \mathbb{E}\Big[\big(G + Ut\big)_-^2 I\{\Omega_\searrow\}\Big].
\end{aligned}
$$

Here, the positive part of any real $a$ is denoted $a_+ = \max(0, a)$, the negative part $a_- = \max(-a, 0)$ and the square is always taken after the positive/negative parts, i.e., $a_+^2 = (a_+)^2$ and $a_-^2 = (a_-)^2$. If $\inf_{t \in \mathbb{R}} \varphi(t) = 0$ then $\delta_\infty$ is interpreted as $\delta_\infty = +\infty$. Above, the infimum over $t \in \mathbb{R}$ always admits a minimizer $t_* \in \mathbb{R}$ under Assumption 2.4 (see Lemma A.1).

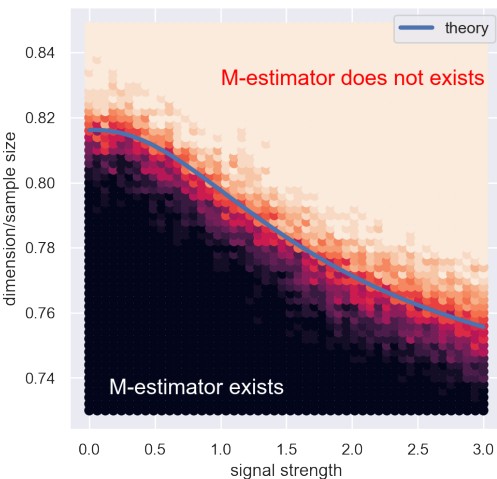

Figure 2. Count of instances where the minimizer in (1) exists for varying $p/n$ and signal strength. Simulation parameter: $n = 1500$, 20 repetitions, $\ell_y(u) = e^u - yu$ is the Poisson loss, $y_i \mid \boldsymbol{x}_i$ satisfies the Poisson model (8).

Our first result is that the threshold $\delta_\infty$ characterizes the phase transition regarding whether the M-estimator exists or not with high-probability under the proportional regime $n/p \to \delta$.

**Theorem 2.6.** *As $n, p \to +\infty$ with $n/p \to \delta$, we have*

$$\mathbb{P}\begin{pmatrix} \textit{The M-estimator exists,} \\ \textit{i.e., the inf (1) is attained} \end{pmatrix} \to \begin{cases} 1 & \textit{if } \delta > \delta_\infty, \\ 0 & \textit{if } \delta < \delta_\infty. \end{cases}$$

Theorem 2.6 is numerically verified by Figure 2 for Poisson regression (see Section 4 for details). The proof of Theorem 2.6 is based on a common argument based on conic geometry and the Gaussian kinematic formula given by Amelunxen et al. (2014). This use of the kinematic formula of Amelunxen et al. (2014) is similar to the argument in Candès & Sur (2020) for the binary logistic regression model.

A more technical question, that requires an investigation beyond the application of the kinematic formula of Amelunxen et al. (2014), is whether the critical threshold also characterizes the phase transition regarding the existence of solution to the nonlinear system (4). A formal proof of existence of a unique solution to (4) is important, as it is required to leverage the Convex Gaussian Minmax Theorem (CGMT) of Thrampoulidis et al. (2018). For instance, the works Salehi et al. (2019); Loureiro et al. (2021) which apply the CGMT in generalized linear models to study $\hat{\boldsymbol{b}}$, assume in their theorems that the system (2) admits a unique solution.

**Theorem 2.7.**

- *If $\delta \le \delta_\infty$, the system (2) has no solution.*

- *If $\delta > \delta_\infty$, the system (2) has a unique solution.*

To our knowledge, a proof of this relationship between the critical threshold $\delta_\infty$ and the existence of a solution to the system (2) is new, even in the case of binary logistic regression. The global null case was handled in (Sur et al., 2019); however, outside the global null case, this phenomenon was observed numerically in (Sur & Candès, 2019) without proof—see the discussion after eq. (16) of the supplement of (Sur & Candès, 2019).

Montanari et al. (2023) generalized the threshold $\delta_\infty$ of Candès & Sur (2020) for linear separation in binary classification (or equivalently for the existence of (12) with the logistic loss), allowing an arbitrary single-index model for $\mathbb{P}(y_i = 1 \mid \boldsymbol{x}_i)$. Montanari et al. (2023) further established the existence of a unique solution to a system governing the behavior of max-margin classifiers for any $\delta < \delta_\infty$, that is, for any $\delta$ such that the data is linearly separable with high-probability. We emphasize that the system studied in Montanari et al. (2023) is different and complementary from the system (2) of interest in the present paper: The system of Montanari et al. (2023) governs the behavior of max-margin classifiers for $\delta < \delta_\infty$ while the system (2) studied here governs the behavior of the M-estimator (1) for which Theorem 2.7 establishes existence of solutions if and only if $\delta > \delta_\infty$.

The tools we use to obtain Theorem 2.7 are based on the existence of a solution to a convex minimization problem in an infinite-dimensional Hilbert space which is the focus of the next section.

## 3. Infinite Dimensional Optimization Problem

In this section, we define the mathematical objects at the heart of the proof of Theorem 2.7 and outline the proof strategy. The key is the analysis of an infinite-dimensional convex optimization problem that is in a dual relationship with the nonlinear system. The use of such infinite-dimensional optimization problems to prove the existence of solutions to nonlinear systems of equations was pioneered in Montanari et al. (2023) and later used in the context of boosting and L1 interpolation (Liang & Sur, 2022), to analyse the Lasso (Celentano et al., 2023), and for robust regularized regression (Bellec & Koriyama, 2023).

The notation and setup of this infinite-dimensional convex optimization problem is heavily inspired by Bellec & Koriyama (2023), which studies the existence of solutions to systems of a similar nature to (2) in robust regression. In the robust regression setup of Bellec & Koriyama (2023) where coercive and Lipschitz loss functions $\ell_{y_i}(\cdot)$ are considered, the corresponding system has always a unique solution and the M-estimator always exists: there is no phase transition. A novelty of the present paper is to explain how these tools,

in particular the infinite-dimensional optimization below, can be used to predict the phase transition for the existence of a minimizer in (1) and the phase transition for the existence of solutions to the system (2).

To begin with, let us consider the almost sure equivalent classes $\mathcal{H}$ of squared integrable measurable functions of $(G, U, Y)$

$$\mathcal{H} = \{v : \mathbb{R}^3 \to \mathbb{R} : \mathbb{E}[v(G, U, Y)^2] < +\infty\},$$

where $G \sim N(0, 1)$ and independent of $(U, Y)$. Here $(U, Y)$ is equal in distribution to $(y_i, \boldsymbol{x}_i^\top \boldsymbol{w})$. Almost sure equivalence classes of $\mathcal{H}$ form a Hilbert space equipped with the usual inner product $\langle u, v \rangle := \mathbb{E}[u(G, U, Y)v(G, U, Y)]$ and corresponding Hilbert norm $\|v\| = \sqrt{\langle v, v \rangle}$. We will sometimes refer to $\mathcal{H}$ itself as the Hilbert space, in this case we implicitly identify random variables $v(G, U, Y)$ that are equal almost surely. For brevity, inside an expectation and probability signs with respect to the probability measure of $(G, U, Y)$, we simply write $v$ to denote the random variable $v(G, U, Y)$. For instance, we write simply $\mathbb{E}[vG]$ instead of $\mathbb{E}[v(G, U, Y)G]$.

Now we define two functions $\mathcal{G}$ and $\mathcal{L}$ as follows:

$$\mathcal{G} : \mathcal{H} \to \mathbb{R}, \quad v \mapsto \|v\| - \mathbb{E}[vG]/\sqrt{1 - \delta^{-1}}$$

and

$$\mathcal{L} : \mathbb{R} \times \mathcal{H} \to \mathbb{R} \cup \{+\infty\},$$
$$(a, v) \mapsto \begin{cases} \mathbb{E}[\ell_Y(aU + v)] & \text{if } \mathbb{E}[|\ell_Y(aU + v)|] < +\infty \\ +\infty & \text{otherwise} \end{cases}$$

Here, $\mathcal{L}$ is a proper lower semicontinuous convex function, while $\mathcal{G}$ is a Lipschitz, finite valued, and convex function (See Lemma B.1 and Lemma B.2). With these functions $(\mathcal{L}, \mathcal{G})$, we claim that the system of nonlinear equations (2) admits a unique solution if and only if the following infinite-dimensional convex optimization problem over $\mathbb{R} \times \mathcal{H}$

$$\min_{(a,v) \in \mathbb{R} \times \mathcal{H}} \mathcal{L}(a, v) \quad \text{subject to} \quad \mathcal{G}(v) \leq 0 \qquad (5)$$

admits a unique minimizer $(a_*, v_*) \in \mathbb{R} \times \mathcal{H}$. We will make this point more precise in the next paragraph.

The key to such an equivalence between the nonlinear system (2) and infinite-dimensional optimization problem (5) is the existence of the Lagrange multiplier associated with the constraint $\mathcal{G}(v) \leq 0$. By Proposition 27.31 of Bauschke & Combettes (2017), an element $(a_*, v_*) \in \mathbb{R} \times \mathcal{H}$ solves the constrained optimization problem $\min_{(a,v):\mathcal{G}(v)\leq 0} \mathcal{L}(v)$ if and only if there exists a Lagrange multiplier $\mu_* \geq 0$ such

that the KKT condition

$$-\mu_* \partial \mathcal{G}(v_*) \cap \partial_v \mathcal{L}(a_*, v_*) \neq \emptyset$$
$$\partial_a \mathcal{L}(a, v) \ni 0$$
$$\mu_* \mathcal{G}(v_*) = 0 \qquad (6)$$
$$\mathcal{G}(v_*) \leq 0$$

is satisfied, where $\partial \mathcal{G}$ and $\partial \mathcal{L}$ are the subdifferentials of the convex functions $\mathcal{G}, \mathcal{L}$. Furthermore, we will argue that the Lagrange multiplier $\mu_*$ is strictly positive. Combined with $\mu_* \mathcal{G}(v_*) = 0$ in the KKT condition (6), this means that $\mathcal{G}(v_*) = 0$, i.e., the constraint $\mathcal{G}(v) \leq 0$ is binding. Following Bellec & Koriyama (2023), equipped with this positive Lagrange multiplier $\mu_* > 0$ and the binding condition $\mathcal{G}(v_*) = 0$, we establish the following equivalence between the minimizer of the optimization problem (5) and the solution to the nonlinear system of equations (2).

**Theorem 3.1** (Equivalence).

- *Suppose $(a_*, v_*) \in \mathbb{R} \times \mathcal{H}$ solves the constrained optimization problem (5) with $\|v_*\| > 0$. Let $\mu_*$ be the Lagrange multiplier satisfying the KKT condition. Let $(\gamma_*, \sigma_*)$ be the positive scalar defined by*

$$\sigma_* = \|v_*\|/\sqrt{1 - \delta^{-1}} > 0, \quad \gamma_* = \|v_*\|/\mu_* > 0.$$

  *Then the pair $(a_*, \sigma_*, \gamma_*)$ solves the nonlinear system of equation.*

- *If $(a_*, \gamma_*, \sigma_*) \in \mathbb{R} \times \mathbb{R}_{>0} \times \mathbb{R}_{>0}$ satisfies the nonlinear system, letting*

$$v_* = \text{prox}[\gamma_* \ell_Y](a_* U + \sigma_* G) - a_* U$$

  *$(a_*, v_*)$ solves the optimization problem (5) with $\|v_*\| = \sigma_* \sqrt{1 - \delta^{-1}} > 0$ and the KKT condition (6) is satisfied for $\mu_* = \sigma_* \sqrt{1 - \delta^{-1}}/\gamma_* > 0$.*

Theorem 3.1 implies that the nonlinear system of equations (2) admits a unique solution $(a_*, \gamma_*, \sigma_*) \in \mathbb{R} \times \mathbb{R}_{>0} \times \mathbb{R}_{>0}$ if and only if the optimization problem $\min_{\mathcal{G}(v)\leq 0} \mathcal{L}(a, v)$ admits a unique solution $(a_*, v_*) \in \mathbb{R} \times \mathcal{H}$ with $v_* \neq 0$ and a unique Lagrange multiplier $\mu_* > 0$ satisfying the KKT condition (6). In order to apply Theorem 3.1, we need to establish that the degenerate case $v_* = 0$ cannot happen.

**Lemma 3.2.** *(Non-degeneracy) If $(a_*, v_*)$ solves the optimization problem (5), then $v_* \neq 0$.*

In the proof of Lemma 3.2, the differentiability of the loss $\ell_y$ is crucial in preventing the degenerate case $v_* = 0$. When the loss is not differentiable, another different threshold, $\delta_{\text{perfect}}$, emerges to determine whether $v_* = 0$ or $v_* \neq 0$ occurs (Bellec & Koriyama, 2023).

**Lemma 3.3.** *(Uniqueness) The minimizer of the optimization problem* (5) *is unique if it exists. Furthermore, the Lagrange multiplier satisfying the KKT condition* (6) *is also unique.*

Combining Theorem 3.1, Lemma 3.2, and Lemma 3.3, we conclude that the system of nonlinear equations has a unique solution if and only if the infinite-dimensional optimization problem admits a minimizer. This equivalence is useful because studying the existence of solutions to the system (2) directly is a tenuous analysis problem that has been solved in only a few cases: for the Lasso (Bayati & Montanari, 2011; Miolane & Montanari, 2021), or for the global null case of logistic regression (Sur et al., 2019). Instead of studying the system (2) directly, this equivalence allows us to focus on the existence of minimizer for the infinite-dimensional convex minimization problem (5). Even though the problem is infinite-dimensional, there is a well-developed theory for convex minimization in Hilbert spaces (Bauschke & Combettes, 2017) which can be leveraged to study (5); including the KKT condition or the fact that a coercive convex objective function admits a minimizer.

It remains to establish that the existence of a minimizer for the optimization problem is governed by the threshold $\delta_\infty$ (which is the same as in Theorem 2.6), thus completing the proof of Theorem 2.7.

**Theorem 3.4.** *Let $\delta_\infty$ be the threshold defined in* (4). *Then we have the following:*

- *If $\delta \leq \delta_\infty$, the problem* (5) *has no minimizer.*

- *If $\delta > \delta_\infty$, the problem* (5) *admits a minimizer.*

Let us explain where this phase transition comes from, starting from the case $\delta \leq \delta_\infty$ where we claim that (5) admits no minimizer. The first idea concerns $\mathcal{L}$: a natural avenue to show that there is no minimizer is to try to find a direction $(a, v)$ such that $t \mapsto \ell_Y(s(aU + v))$ is decreasing in $s > 0$ for all realizations of $(Y, U)$ (we are looking for such a direction because if a convex function admits a ray along which it is decreasing, then it admits no minimizer). By considering the three events $\Omega_\vee(Y), \Omega_\nearrow(Y), \Omega_\searrow(Y)$, this motivates the definition of the cone $\mathcal{C} \subset \mathbb{R} \times \mathcal{H}$ defined as

$$(a, p) \in \mathcal{C} \Leftrightarrow aU + p \begin{cases} \leq 0 & \text{under } \Omega_\nearrow(Y) \\ = 0 & \text{under } \Omega_\vee(Y) \\ \geq 0 & \text{under } \Omega_\searrow(Y). \end{cases} \quad (7)$$

Next, the direction we are looking for should also satisfy the constraint $\mathcal{G}(v) \leq 0$. This motivates the consideration of

$$(a_*, p_*) \in \text{argmin}_{(a,p) \in \mathcal{C}} \mathbb{E}[(G - p)^2]$$

because among all $(a, p) \in \mathcal{C}$ such that $\|p\| = \|p^*\|$, the $p^*$ defined above necessarily has larger correlation $\mathbb{E}[Gp^*]$

with $G$, so $p^*$ has a better chance to satisfy the constraint $\mathcal{G}(p^*) \leq 0$ than $p$. We show in Lemma A.1 that

$$\|p_*\| = \sqrt{1 - \delta_\infty^{-1}}, \qquad \|p_*\|^2 = \mathbb{E}[Gp^*].$$

This immediately gives that if $\delta \leq \delta_\infty$ then $p^*$ satisfies the constraint $\mathcal{G}(p_*) \leq 0$ in (5) (this is serendipitous and "barely" works out, since for any $\delta > \delta_\infty$ the constraint would be violated). If $\delta \leq \delta_\infty$, we have exhibited a direction $(a_*, p_*)$ such that $\mathcal{L}(sa_*, sp_*)$ is decreasing in $s > 0$ and such that $\mathcal{G}(sp_*) = s\mathcal{G}(p_*) \leq 0$. Since we have found a direction along which the objective function is decreasing and which satisfies the constraint, the minimization problem admits no minimizer. All these arguments are made precise and formally proved in the appendix.

For the other side of the phase transition, $\delta > \delta_\infty$, the following idea is used, which exhibits a similarly serendipitous phenomenon. Here we must show that (5) admits a minimizer. This is typically obtained by showing that the objective function is coercive (i.e., has bounded level sets): that any $v, a$ satisfying the constraint such that $\mathcal{L}(a, v) \leq \xi$ for some $\xi \in \mathbb{R}$ must satisfy $|a| + \|v\| \leq C(\xi)$ for some constant $C(\xi)$. We break the problem by breaking $v$ into two parts, $v = \tilde{v} + (v - \tilde{v})$ where

$$\tilde{v} = -aU + \begin{cases} (aU + v)_- & \text{under } \Omega_\nearrow(Y) \\ (aU + v)_+ & \text{under } \Omega_\searrow(Y) \\ 0 & \text{under } \Omega_\vee(Y) \end{cases}$$

so that $(a, \tilde{v}) \in \mathcal{C}$ for all $(a, v) \in \mathbb{R} \times \mathcal{H}$. Here, $aU + \tilde{v}$ is the additive part of $aU + v$ that satisfies the constraints in the definition of $\mathcal{C}$ and carries a risk of generating a ray along which the objective function is decreasing (as for $p_*$ in the previous paragraph). On the other hand, the other additive part $v - \tilde{v}$ can be bounded using the one-sided coercivity of $\ell_Y$ in $\Omega_\nearrow(Y)$ or $\Omega_\searrow(Y)$, and the two-sided coercivity under $\Omega_\vee(Y)$ (this is made precise and formally proved in the appendix, see Lemma E.2). To bound $\tilde{v}$ (or directly $v$) after having controlled $v - \tilde{v}$, we establish using the properties of $p_*$ the inequality

$$\|v\|(\sqrt{1 - \delta^{-1}} - \|p_*\|) \leq \mathbb{E}[v(G - p_*)]$$
$$\leq \mathbb{E}[(\tilde{v} - v)(p_* - G)],$$

see (27) in the appendix. The factor $(\sqrt{1 - \delta^{-1}} - \|p_*\|)$ in the left-hand side is positive only on the side $\delta > \delta_\infty$ of the phase transition, which serendipitously lets us prove that $\|v\|$ is in turn bounded, that the objective function is (5) is coercive, that (5) consequently admits a minimizer, and by the equivalence in Theorem 3.1 that the system (5) admits a solution. This strategy is made precise and formally proved in Appendix E.

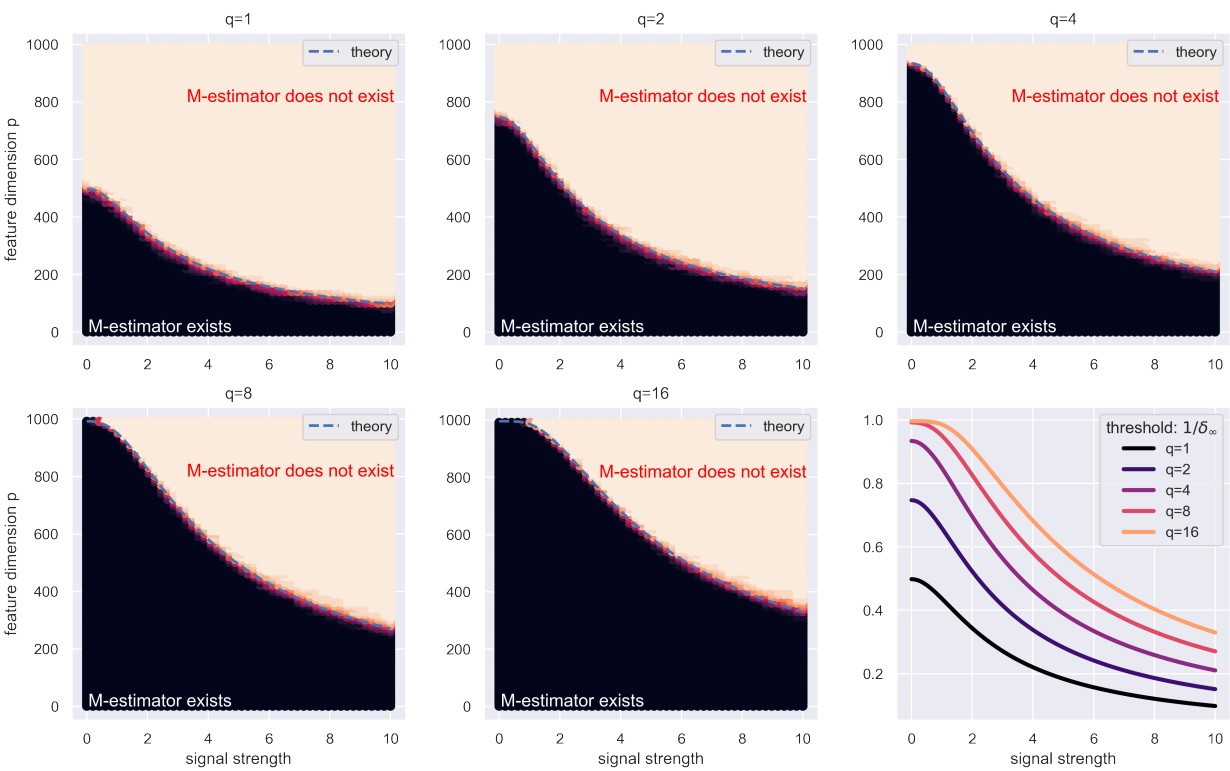

*Figure 3.* Count of instances where the minimizer in (1) exists for varying $p/n$ and signal strength $\kappa$. Simulation parameter: $n = 1000$, 20 repetitions, $y_i \mid \boldsymbol{x}_i \sim$ satisfies the binomial model $\text{Binomial}(q, p_i)$ as in (9).

## 4. Numerical Simulation

We generate the covariates $(\boldsymbol{x}_i)_{i=1}^{n} \overset{\text{iid}}{\sim} N(\boldsymbol{0}_p, I_p)$ and responses $y_i \mid \boldsymbol{x}_i$ according to the Poisson model

$$\forall k \in \mathbb{N}, \quad \mathbb{P}\Big(y_i = k \mid \boldsymbol{x}_i\Big) = \frac{\lambda_i^k}{k!} \exp(-\lambda_i), \quad (8)$$

where $\lambda_i = \exp(-\kappa \boldsymbol{e}_1^\top \boldsymbol{x}_i)$. Here, $\boldsymbol{e}_1$ is the first canonical basis vector, and $\kappa \geq 0$ is the signal strength. We fix $n = 1000$ and for varying values of $(p/n, \kappa)$, we generate 20 datasets of $(\boldsymbol{x}_i, y_i)_{i=1}^{n}$. For each dataset $(\boldsymbol{x}_i, y_i)_{i=1}^{n}$, we solve the optimization problem $\inf_{\boldsymbol{b} \in \mathbb{R}^p} \sum_{i=1}^{n} \ell_{y_i}(\boldsymbol{x}_i^\top \boldsymbol{b})$ using the Poisson loss $\ell_{y_i}(t) = \exp(t) - y_i t$ and record whether a minimizer exists using linear programming. In Figure 2, we normalize the count of instances where a minimizer exists by dividing by 20, with the black points indicating higher rates of existence. Additionally, we plot the theoretical threshold $1/\delta_\infty$ defined in (4) and compare it with the empirical result. The theoretical threshold effectively separates the two regions, delineating where the M-estimator exists and where it does not.

Next, generate $y_i \mid \boldsymbol{x}_i$ according to the Binomial distribution

$$\forall k \in [q], \quad \mathbb{P}(y_i = k \mid \boldsymbol{x}_i) = \binom{q}{k} p_i^k (1 - p_i)^{q-k}, \quad (9)$$

where $p_i = \frac{1}{1 + \exp(-\kappa \boldsymbol{e}_1^\top \boldsymbol{x}_i)}$. Here, $q \in \{1, 2, \dots\} = \mathbb{N}$ is the number of measurement, a hyperparameter to be specified. Given the data set $(\boldsymbol{x}_i, y_i)_{i=1}^{n}$, we solve the optimization problem $\inf_{\boldsymbol{b} \in \mathbb{R}^p} \sum_{i=1}^{n} \ell_{y_i}(\boldsymbol{x}_i^\top \boldsymbol{b})$ using the loss $\ell_y(t) = q \log(1 + \exp(t)) - yt$; in other words, we compute the corresponding MLE. Similarly to Figure 2, Figure 3 plots the count of instances where a minimizer exists in (1) along with the theoretical threshold $1/\delta_\infty$ for each hyperparameter $q \in \{1, 2, 4, 8, 16\}$. When $q = 1$, the simulation setting is reduced to the Binary logistic regression, thereby recovering the figure in (Candès & Sur, 2020). The result for $q \geq 2$ is new, and we observe that the generalized threshold (4) predicts well the existence of the MLE.

## Impact Statement

This paper presents work whose goal is to advance the field of Machine Learning. There are many potential societal consequences of our work, none which we feel must be specifically highlighted here.

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

## A. Derivation of threshold from convex geometry

Define $\delta_\infty \in (0, +\infty]$ as

$$1/\delta_\infty := \inf_{t \in \mathbb{R}} \varphi(t)$$

where $\varphi : \mathbb{R} \to \mathbb{R}$ is the convex function defined as

$$\varphi(t) := \mathbb{E}\big[(G + Ut)^2 I\{\Omega_\vee(Y)\}\big] + \mathbb{E}\big[(G + Ut)_+^2 I\{\Omega_\nearrow(Y)\}\big] + \mathbb{E}\big[(G + Ut)_-^2 I\{\Omega_\searrow(Y)\}\big].$$

Here $\Omega_\vee, \Omega_\nearrow, \Omega_\searrow$ are $\sigma(Y)$-measurable events defined in (3). We denote by $\mathcal{H}$ the Hilbert space which consists of measurable function of $(G, Y, U)$ with finite second moments. Let $\mathcal{C} \subset \mathbb{R} \times \mathcal{H}$ be the cone defined as

$$(t, p) \in \mathcal{C} \Leftrightarrow tU + p \begin{cases} \leq 0 & \text{under } \Omega_\nearrow(Y) \\ = 0 & \text{under } \Omega_\vee(Y) \\ \geq 0 & \text{under } \Omega_\searrow(Y) \end{cases} \tag{10}$$

**Lemma A.1.** *The threshold $\delta_\infty$ can be represented as*

$$\delta_\infty^{-1} = \inf_{t \in \mathbb{R}} \varphi(t) = \inf_{t \in \mathbb{R}} \mathbb{E}[(G - p(t))^2] = \inf_{(t,p) \in \mathcal{C}} \mathbb{E}[(G - p)^2]$$

*where $(t, p(t)) \in \mathcal{C}$ for all $t \in \mathbb{R}$ and $p(t)$ is given by*

$$p(t) := -tU + \begin{cases} 0 & \text{under } \Omega_\vee(Y) \\ (G + Ut)_- & \text{under } \Omega_\nearrow(Y) \\ (G + Ut)_+ & \text{under } \Omega_\searrow(Y). \end{cases}$$

*Suppose that the law of $(U, Y)$ satisfies*

$$\mathbb{E}\Big[U^2\big(I\{\Omega_\vee(Y)\} + I\{\Omega_\searrow(Y), U > 0\} + I\{\Omega_\nearrow(Y), U < 0\}\big)\Big] > 0,$$
$$\mathbb{E}\Big[U^2\big(I\{\Omega_\vee(Y)\} + I\{\Omega_\searrow(Y), U < 0\} + I\{\Omega_\nearrow(Y), U > 0\}\big)\Big] > 0. \tag{11}$$

*Then the map $\varphi$ is coercive, i.e., $\lim_{|t| \to +\infty} \varphi(t) = +\infty$ and $\inf_{t \in \mathbb{R}} \varphi(t)$ admits a minimizer $t_* \in \mathbb{R}$. Furthermore, the optimal $p_* = p(t_*) \in \mathcal{H}$ satisfies*

$$\mathbb{E}[p_*^2] = \mathbb{E}[p_* G], \quad \|p_*\| = \sqrt{1 - \delta_\infty^{-1}}.$$

*Proof.* Fix $t \in \mathbb{R}$. By the definition of the cone $\mathcal{C}$ and $\varphi$, it easily follows that

$$\inf_{p \in \mathcal{H} : (t,p) \in \mathcal{C}} \mathbb{E}[(G - p)^2] = \mathbb{E}[(G - p(t))^2] = \varphi(t)$$

This proves the representation $\inf_{t \in \mathbb{R}} \varphi(t) = \inf_{(t,p) \in \mathcal{C}} \mathbb{E}[(G - p)^2] = \inf_{t \in \mathbb{R}} \mathbb{E}[(G - p(t))^2]$.

Next, let us show that the map $\varphi : \mathbb{R} \to \mathbb{R}$

$$\varphi(t) = \mathbb{E}\big[(G + Ut)^2 I\{\Omega_\vee(Y)\}\big] + \mathbb{E}\big[(G + Ut)_+^2 I\{\Omega_\nearrow(Y)\}\big] + \mathbb{E}\big[(G + Ut)_-^2 I\{\Omega_\searrow(Y)\}\big]$$

is coercive. Since $t \mapsto \varphi(t)$ is convex, it suffices to show the coercivity, i.e., $\lim_{|t| \to +\infty} \varphi(t) = +\infty$. For the first term, expanding the square of $(G + Ut)^2$, it immediately follows that

$$\mathbb{E}[(G + Ut)^2 I\{\Omega_\vee(Y)\}] = t^2 \mathbb{E}[U^2 I\{\Omega_\vee(Y)\}] + O(t)$$

For the second term $\mathbb{E}[(G + Ut)_+^2 I\{\Omega_\nearrow(Y)\}]$,

$$\mathbb{E}[(G + Ut)_+^2 I\{\Omega_\nearrow(Y)\}] = \mathbb{E}[(G + Ut)^2 I\{\Omega_\nearrow(Y), \, G + Ut > 0\}]$$
$$= t^2 \mathbb{E}[U^2 I\{\Omega_\nearrow(Y)\} I\{G + Ut > 0\}] + O(|t|)$$

as $|t| \to +\infty$. By $\mathbb{E}[U^2] = 1 < +\infty$, the dominated convergence theorem implies

$$
\begin{aligned}
\mathbb{E}[U^2 I\{\Omega_\nearrow(Y)\} I\{G + Ut > 0\}] &= \mathbb{E}[U^2 I\{\Omega_\nearrow(Y)\} I\{G + Ut > 0, U > 0\}] \\
&\quad + \mathbb{E}[U^2 I\{\Omega_\nearrow(Y)\} I\{G + Ut > 0, U < 0\}] \\
&\to \begin{cases} \mathbb{E}[U^2 I\{\Omega_\nearrow(Y), U > 0\}] & t \to +\infty \\ \mathbb{E}[U^2 I\{\Omega_\nearrow(Y), U < 0\}] & t \to -\infty \end{cases}
\end{aligned}
$$

Thus,

$$
\frac{1}{t^2}\, \mathbb{E}[(G + Ut)_+^2 I\{\Omega_\nearrow(Y)\}] \to \begin{cases} \mathbb{E}[U^2 I\{\Omega_\nearrow(Y), U > 0\}] & t \to +\infty \\ \mathbb{E}[U^2 I\{\Omega_\nearrow(Y), U < 0\}] & t \to -\infty \end{cases}
$$

By the same argument, we have

$$
\frac{1}{t^2}\, \mathbb{E}[(G + Ut)_-^2 I\{\Omega_\searrow(Y)\}] \to \begin{cases} \mathbb{E}[U^2 I\{\Omega_\searrow(Y), U < 0\}] & t \to +\infty \\ \mathbb{E}[U^2 I\{\Omega_\searrow(Y), U > 0\}] & t \to -\infty \end{cases}
$$

Putting them together, we obtain

$$
\frac{\varphi(t)}{t^2} \to \begin{cases} \mathbb{E}[U^2 I\{\Omega_\vee(Y)\}] + \mathbb{E}[U^2 I\{\Omega_\nearrow(Y), U > 0\}] + \mathbb{E}[U^2 I\{\Omega_\searrow(Y), U < 0\}] & t \to +\infty \\ \mathbb{E}[U^2 I\{\Omega_\vee(Y)\}] + \mathbb{E}[U^2 I\{\Omega_\nearrow(Y), U < 0\}] + \mathbb{E}[U^2 I\{\Omega_\searrow(Y), U > 0\}] & t \to -\infty \end{cases}.
$$

Since the limit is strictly positive by the condition (11), the map $t \mapsto \varphi(t)$ is coercive, and hence $\inf_{t \in \mathbb{R}} \varphi(t)$ is attained at some $t = t_* \in \mathbb{R}$.

Finally, we prove $\mathbb{E}[p_*^2] = \mathbb{E}[p_* G]$ and $\|p_*\| = \sqrt{1 - \delta_\infty^{-1}}$. The stationary condition of $\varphi'(t_*) = 0$ gives

$$
\begin{aligned}
0 &= 2\,\mathbb{E}[(G + Ut_*)U I\{\Omega_\vee(Y)\}] + 2\,\mathbb{E}[(G + Ut_*)_+ U I\{\Omega_\nearrow(Y)\}] + 2\,\mathbb{E}[(G + Ut_*)_- U I\{\Omega_\searrow(Y)\}] \\
&= 2\,\mathbb{E}[(G - p_*)U]
\end{aligned}
$$

where

$$
p_* := p(t_*) = -t_* U + (G + Ut_*)_- I\{\Omega_\nearrow(Y)\} + (G + Ut_*)_+ I\{\Omega_\searrow(Y)\}
$$

Here, $\mathbb{E}[GU] = 0$ since $G$ and $U$ are independent standard normal. Thus, the last equation gives $\mathbb{E}[p_* U] = 0$. Then, we have

$$
\begin{aligned}
\mathbb{E}[p_*^2] &= -t_*^2 + t_*^2 + \mathbb{E}[p_*^2] + 2t_*\,\mathbb{E}[p_* U] && \mathbb{E}[p_* U] = 0 \\
&= -t_*^2 + \mathbb{E}[(p_* + Ut_*)^2] \\
&= -t_*^2 + \mathbb{E}\left[I\{\Omega_\nearrow(Y)\}(G + Ut_*)_-^2\right] + \mathbb{E}\left[I\{\Omega_\searrow(Y)\}(G + Ut_*)_+^2\right], \\
\mathbb{E}[p_* G] &= \mathbb{E}[(p_* + Ut_*)G] && \mathbb{E}[UG] = 0 \\
&= \mathbb{E}[(t_* U + G)_- G I\{\Omega_\nearrow(Y)\}] + \mathbb{E}[(t_* U + G)_+ G I\{\Omega_\searrow(Y)\}]
\end{aligned}
$$

By $(G + t_* U)_\pm^2 = (G + t_* U)_\pm (G + t_* U)$ and the definition of $p_*$, we get

$$
\begin{aligned}
&\mathbb{E}[p_*^2] - \mathbb{E}[p_* G] \\
&= -t_*^2 + \mathbb{E}[(t_* U + G)_- t_* U I\{\Omega_\nearrow(Y)\}] + \mathbb{E}[(t_* U + G)_+ t_* U I\{\Omega_\searrow(Y)\}] \\
&= -t_*^2 + t_*\,\mathbb{E}\left[U\left((G + Ut_*)_- I\{\Omega_\nearrow(Y)\} + (G + Ut_*)_+ I\{\Omega_\searrow(Y)\}\right)\right] \\
&= -t_*^2 + t_*\,\mathbb{E}[U(t_* U + p_*)] && \text{by the definition of } p_* \\
&= -t_*^2 + t_*^2 + 0 && \mathbb{E}[U^2] = 1,\ \mathbb{E}[U p_*] = 0 \\
&= 0.
\end{aligned}
$$

The equation $\|p_*\| = \sqrt{1 - \delta_\infty^{-1}}$ follows from $\|p_*\|^2 = \mathbb{E}[p_* G]$ and $\delta_\infty^{-1} = \mathbb{E}[(G - p_*)^2]$.  $\square$

### A.1. Proof of Theorem 2.6

**Lemma A.2.** *Suppose that $\ell_{y_i}$ is strictly convex, $C^1$, and $\inf_u \ell_{y_i}(u)$ is finite. Then the M-estimator does not exist if and only if*

$$\exists \boldsymbol{b}_* \in \mathbb{R}^p \setminus \{0\} \text{ such that } \left( \forall i \in [n], \quad \boldsymbol{x}_i^\top \boldsymbol{b}_* \begin{cases} \geq 0 & under\ \Omega_\searrow(y_i) \\ = 0 & under\ \Omega_\vee(y_i) \\ \leq 0 & under\ \Omega_\nearrow(y_i) \end{cases} \right) \tag{12}$$

*Proof.* (12) **holds $\Rightarrow$ M-estimator does not exist:**. Let $L(\boldsymbol{b}) := \sum_{i=1}^n \ell_{y_i}(\boldsymbol{x}_i^\top \boldsymbol{b})$ be the objective function. Suppose there exists $\boldsymbol{b}_* \in \mathbb{R}^p \setminus \{\boldsymbol{0}_p\}$ such that (12) is satisfied. Then, the map $\mathbb{R}_{\geq 0} \ni t \mapsto L(t\boldsymbol{b}_*)$ is uniformly bounded by $\sum_{i=1}^n \ell_{y_i}(0)$. For all $\boldsymbol{b} \in \mathbb{R}^p$, let $\boldsymbol{b}_\nu$ be the convex combination

$$\boldsymbol{b}_\nu = (1 - 1/\nu)\boldsymbol{b} + (1/\nu)(\nu \boldsymbol{b}_*), \quad \nu > 1.$$

Note $\boldsymbol{b}_\nu \to \boldsymbol{b} + \boldsymbol{b}_*$ as $\nu \to +\infty$. By the convexity of $L(\boldsymbol{b})$ and uniform bound $\sup_{t \geq 0} L(t\boldsymbol{b}_*) \leq \sum_{i=1} \ell_{y_i}(0)$, we have

$$L(\boldsymbol{b}_\nu) \leq (1 - 1/\nu)L(\boldsymbol{b}_*) + (1/\nu)L(\nu \boldsymbol{b}_*) \leq (1 - 1/\nu)L(\boldsymbol{b}_*) + (1/\nu) \sum_{i=1}^n \ell_{y_i}(0)$$

Taking $\nu \to +\infty$, the RHS converges to $L(\boldsymbol{b}_*)$, while the LHS converges to $L(\boldsymbol{b} + \boldsymbol{b}_*)$ by the continuity of $L(\cdot)$ and $\boldsymbol{b}_\nu \to \boldsymbol{b} + \boldsymbol{b}_*$, so we are left with

$$L(\boldsymbol{b} + \boldsymbol{b}_*) \leq L(\boldsymbol{b}), \quad \forall \boldsymbol{b} \in \mathbb{R}^p.$$

Now we suppose that the M-estimator $\hat{\boldsymbol{b}} \in \arg\min_{\boldsymbol{b} \in \mathbb{R}^p} L(\boldsymbol{b})$ exists. Setting $\boldsymbol{b} = \hat{\boldsymbol{b}}$ in the above display, since $\boldsymbol{b}_* \neq \boldsymbol{0}_p$, we know that $\tilde{\boldsymbol{b}} := \boldsymbol{b}_* + \hat{\boldsymbol{b}}$ is also a minimizer with $\tilde{\boldsymbol{b}} \neq \hat{\boldsymbol{b}}$. For all $t \in (0, 1)$, by the convexity of $L(\cdot)$, we have

$$L(t\hat{\boldsymbol{b}} + (1 - t)\tilde{\boldsymbol{b}}) \leq tL(\hat{\boldsymbol{b}}) + (1 - t)L(\hat{\boldsymbol{b}})$$

Since $\hat{\boldsymbol{b}}$ and $\tilde{\boldsymbol{b}}$ minimize $L$, this holds in equality. With $L(b) = \sum_i \ell_{y_i}(\boldsymbol{x}_i^\top b)$ and by the convexity of $\ell_{y_i}$, the equality condition reads to

$$\forall i \in [n], \quad \ell_{y_i}(t\boldsymbol{x}_i^\top \hat{\boldsymbol{b}} + (1 - t)\boldsymbol{x}_i^\top \tilde{\boldsymbol{b}}) = t\ell_{y_i}(\boldsymbol{x}_i^\top \hat{\boldsymbol{b}}) + (1 - t)\ell_{y_i}(\boldsymbol{x}_i^\top \tilde{\boldsymbol{b}})$$

for all $t \in (0, 1)$. By the strict convexity of $\ell_{y_i}$, we must have $\boldsymbol{x}_i^\top \hat{\boldsymbol{b}} = \boldsymbol{x}_i^\top \tilde{\boldsymbol{b}}$ for all $i \in [n]$, i.e., $\hat{\boldsymbol{b}} - \tilde{\boldsymbol{b}} \in \mathrm{Ker}(\boldsymbol{X})$. However, since $\boldsymbol{X} \in \mathbb{R}^{n \times p}$ has iid $N(0, 1)$ entry, $\boldsymbol{X}$ is an $n \times p$ matrix with $\mathrm{rank}(X) = p < n$. This implies $\mathrm{Ker}(\boldsymbol{X}) = \{\boldsymbol{0}_p\}$ and $\hat{\boldsymbol{b}} - \tilde{\boldsymbol{b}} = \boldsymbol{0}_p$, which is a contradiction with $\hat{\boldsymbol{b}} \neq \tilde{\boldsymbol{b}}$. Thus, if (12) holds then the M-estimator does not exist.

**M-estimator does not exist $\Rightarrow$ (12) holds:** Suppose that the M-estimator does not exist. Then, there exists a sequence $(\boldsymbol{b}_k)_{k=1}^\infty$ such that as $k \to +\infty$, we have $L(\boldsymbol{b}_k) \to \inf_{\boldsymbol{b} \in \mathbb{R}^p} L(\boldsymbol{b})$ and $\|\boldsymbol{b}_k\| \to +\infty$. By the compactness of the unit sphere in $\mathbb{R}^p$, we can extract a subsequence $(\boldsymbol{b}'_k)_{k=1}^\infty$ of $(\boldsymbol{b}_k)_{k=1}^\infty$ such that $\boldsymbol{b}'_k/\|\boldsymbol{b}'_k\| \to \boldsymbol{v}$ for a unit sphere vector $\boldsymbol{v} \in \mathbb{R}^p$. Therefore, we can assume without loss of generality that $\boldsymbol{b}_k/\|\boldsymbol{b}_k\|$ converges to a unit sphere vector $\boldsymbol{v}$.

We proceed by contradiction; suppose (12) is not satisfied. Then we can find an index $i = i(\boldsymbol{v}) \in [n]$ associated with the unit sphere vector $\boldsymbol{v}$ such that

$$\boldsymbol{x}_i^\top \boldsymbol{v} \begin{cases} < 0 & under\ \Omega_\searrow(y_i) \\ \neq 0 & under\ \Omega_\vee(y_i) \\ > 0 & under\ \Omega_\nearrow(y_i) \end{cases}$$

If $\Omega_\searrow(y_i)$, or $\Omega_\nearrow(y_i)$, then the derivative of $t \mapsto \ell_{y_i}(t\boldsymbol{x}_i^\top \boldsymbol{v})$ is positive for all $t \geq 1$. If $\Omega_\vee$, it is not necessarily positive right away at $t = 1$, but eventually positive for $t$ large enough, say some $t_*$: for any $t > t_*$, the derivative of $t \mapsto \ell_i(t\boldsymbol{x}_i^\top \boldsymbol{v})$ is positive. Call this derivative at $t_*$, say $A = \boldsymbol{x}_i^\top \boldsymbol{v} \ell'_{y_i}(t_* \boldsymbol{x}_i^\top \boldsymbol{v})$ and $A > 0$.

Let $\boldsymbol{v}_k = \boldsymbol{b}_k/\|\boldsymbol{b}_k\|$ so that $\boldsymbol{x}_i^\top \boldsymbol{v}_k \to \boldsymbol{x}_i^\top \boldsymbol{v}$. Since $\ell_{y_i}$ is $C^1$, for $k$ large enough we have that the derivative of $t \mapsto \ell_{y_i}(t\boldsymbol{x}_i^\top \boldsymbol{v}_k)$ at $t_*$ is larger than $A/2$. Call this derivative $A_k = \boldsymbol{x}_i^\top \boldsymbol{v}_k \ell'_{y_i}(t_* \boldsymbol{x}_i^\top \boldsymbol{v}_k)$ so that $A_k > A/2 > 0$.

By the convexity of $t \mapsto \ell_{y_i}(t\boldsymbol{x}_i^\top \boldsymbol{v}_k)$, we have

$$\ell_{y_i}(\boldsymbol{x}_i^\top \boldsymbol{b}_k) = \ell_{y_i}(\|\boldsymbol{b}_k\|_2 \boldsymbol{x}_i^\top \boldsymbol{v}_k) \geq \ell_{y_i}(t_* \boldsymbol{x}_i^\top \boldsymbol{v}_k) + (\|\boldsymbol{b}\|_k - t_*)A_k \geq \ell_{y_i}(t_* \boldsymbol{x}_i^\top \boldsymbol{v}_k) + (\|\boldsymbol{b}_k\| - t_*)A/2.$$

This gives

$$L(\boldsymbol{b}_k) = \sum_{j=1}^n \ell_{y_j}(\boldsymbol{x}_j^\top \boldsymbol{b}_k) \geq \sum_{j \neq i} \inf_u \ell_{y_j}(u) + \ell_{y_i}(t_* \boldsymbol{x}_i^\top \boldsymbol{v}_k) + (\|\boldsymbol{b}_k\| - t_*)A,$$

where the RHS goes to $\infty$ since $\|\boldsymbol{b}_k\| \to +\infty$ and $A > 0$. This is a contradiction with $L(\boldsymbol{b}_k) \to \inf_{\boldsymbol{b}} L(\boldsymbol{b}) < +\infty$. Thus, if the M-estimator does not exist then (12) holds. $\qquad\square$

*Proof of Theorem 2.6.* By the rotational invariance of $\boldsymbol{x}_i \sim N(\boldsymbol{0}_p, I_p)$, we can assume without loss of generality that $y_i$ depend on $\boldsymbol{x}_i = (x_{i1}, x_{i2}, \ldots, x_{ip}) \in \mathbb{R}^p$ through its first coordinate $x_{i1}$, i.e.,

$$(y_i, x_{i1}) \perp\!\!\!\perp (x_{i2}, \cdots x_{ip}). \tag{13}$$

By Lemma A.2 the M-estimator does not exist if and only if

$$\mathrm{Span}(X\boldsymbol{e}_2, \ldots, X\boldsymbol{e}_p) \cap \mathcal{C}(X\boldsymbol{e}_1, \boldsymbol{y}) \neq \{\boldsymbol{0}_n\},$$

where $\boldsymbol{e}_i (i = 1, 2, \ldots, p)$ are canonical basis vector in $\mathbb{R}^p$ and $\mathcal{C}(X\boldsymbol{e}_1, \boldsymbol{y})$ is the cone in $\mathbb{R}^n$ defined as follows:

$$\forall \boldsymbol{u}, \boldsymbol{y} \in \mathbb{R}^n, \quad \boldsymbol{p} \in \mathcal{C}(\boldsymbol{u}, \boldsymbol{y}) \Leftrightarrow \exists t \in \mathbb{R} \text{ such that } \left( \forall i \in [n], \ tu_i + p_i = \begin{cases} \geq 0 & \text{under } \Omega_\searrow(y_i) \\ = 0 & \text{under } \Omega_\vee(y_i) \\ \leq 0 & \text{under } \Omega_\nearrow(y_i) \end{cases} \right).$$

Here, $\mathrm{Span}(X\boldsymbol{e}_2, \ldots, X\boldsymbol{e}_p)$ is the linear space spanned by the $(p-1)$ vectors $(X\boldsymbol{e}_i)_{i=2}^p$, which is a rotationally invariant random subspace in $\mathbb{R}^n$ with dimension $p-1$ since $X\boldsymbol{e}_i$ are independent standard normal. Furthermore, (13) implies that $\mathrm{Span}(X\boldsymbol{e}_2, \ldots, X\boldsymbol{e}_p)$ is independent of the cone $\mathcal{C}(X\boldsymbol{e}_1, \boldsymbol{y})$. Below, we write $X\boldsymbol{e}_1 = \boldsymbol{u}$ for simplicity. Then, by Theorem I of Amelunxen et al. (2014) we have that for $\eta \in (0, 1/2)$, conditionally on $\boldsymbol{u}, \boldsymbol{y}$,

$$\begin{aligned} p - 1 + \mathrm{stat.\,dim}(\mathcal{C}(\boldsymbol{u}, \boldsymbol{y})) \geq n + n^{1/2+\eta} &\Rightarrow \mathbb{P}(\text{M-estimator exists} \mid \boldsymbol{u}, \boldsymbol{y}) \to 0, \\ p - 1 + \mathrm{stat.\,dim}(\mathcal{C}(\boldsymbol{u}, \boldsymbol{y})) \leq n - n^{1/2+\eta} &\Rightarrow \mathbb{P}(\text{M-estimator exists} \mid \boldsymbol{u}, \boldsymbol{y}) \to 1 \end{aligned} \tag{14}$$

almost everywhere. Here $\mathrm{stat.\,dim}(\mathcal{C}(\boldsymbol{u}, \boldsymbol{y}))$ is the statistical dimension of the cone given by

$$\mathrm{stat.\,dim}(\mathcal{C}(\boldsymbol{u}, \boldsymbol{y})) = n - \mathbb{E}[\mathrm{dist}(\boldsymbol{g}, \mathcal{C}(\boldsymbol{u}, \boldsymbol{y}))^2 \mid X\boldsymbol{e}_1, \boldsymbol{y}] \quad \text{with} \quad \boldsymbol{g} \sim N(\boldsymbol{0}_n, I_n),$$

where the expectation is taken with respect to $\boldsymbol{g}$, which is independent of $(\boldsymbol{u}, \boldsymbol{y})$. Substituting this to (14) with $\eta$ set to $1/4$, we get

$$\begin{aligned} p - 1 \geq \mathbb{E}[\mathrm{dist}(\boldsymbol{g}, \mathcal{C}(\boldsymbol{u}, \boldsymbol{y}))^2 \mid \boldsymbol{u}, \boldsymbol{y}] + n^{3/4} &\Rightarrow \mathbb{P}(\text{M-estimator exists} \mid \boldsymbol{u}, \boldsymbol{y}) \to 0 \\ p - 1 \leq \mathbb{E}[\mathrm{dist}(\boldsymbol{g}, \mathcal{C}(\boldsymbol{u}, \boldsymbol{y}))^2 \mid \boldsymbol{u}, \boldsymbol{y}] - n^{3/4} &\Rightarrow \mathbb{P}(\text{M-estimator exists} \mid \boldsymbol{u}, \boldsymbol{y}) \to 1 \end{aligned}$$

With $p/n \to \delta^{-1}$, if we prove the convergence

$$n^{-1} \mathbb{E}[\mathrm{dist}(\boldsymbol{g}, \mathcal{C}(\boldsymbol{u}, \boldsymbol{y}))^2 \mid \boldsymbol{u}, \boldsymbol{y}] \to^p \delta_\infty^{-1}$$

then we complete the proof. Below we prove this. Recall $\boldsymbol{u} = X\boldsymbol{e}_1$ so that $(\boldsymbol{u}, \boldsymbol{y}) = (u_i, y_i)_{i=1}^n \overset{\mathrm{iid}}{\sim} (U, Y)$. By the explicit gradient identities (B.7)-(B.9) in (Amelunxen et al., 2014), the Euclidean norm of the gradient of $\boldsymbol{g} \mapsto \mathrm{dist}(\boldsymbol{g}, \mathcal{C})^2$ is bounded by $2\|\boldsymbol{g}\|_2$. Thus, conditionally on $\boldsymbol{u}, \boldsymbol{y}$, the Gaussian Poincaré inequality (cf. Theorem 3.20(Boucheron et al., 2013)) yields

$$\mathbb{E}[\mathrm{dist}(\boldsymbol{g}, \mathcal{C}(\boldsymbol{u}, \boldsymbol{y}))^2 \mid \boldsymbol{u}, \boldsymbol{y}] = \mathrm{dist}(\boldsymbol{g}, \mathcal{C}(\boldsymbol{u}, \boldsymbol{y}))^2 + O_P(\sqrt{n}).$$

Here $\text{dist}(\boldsymbol{g}, \mathcal{C}(\boldsymbol{u}, \boldsymbol{y}))^2 = \inf_{\boldsymbol{p} \in \mathcal{C}(\boldsymbol{u}, \boldsymbol{y})} \|\boldsymbol{g} - \boldsymbol{p}\|^2$ is equal to the optimal value of

$$\inf_{(t, \boldsymbol{p}) \in \mathbb{R} \times \mathbb{R}^n} \sum_{i=1}^n (g_i - p_i)^2 \quad \text{subject to} \quad \left( \forall i \in [n], \ tu_i + p_i = \begin{cases} \geq 0 & \text{under } \Omega_{\searrow}(y_i) \\ = 0 & \text{under } \Omega_{\vee}(y_i) \\ \leq 0 & \text{under } \Omega_{\nearrow}(y_i) \end{cases} \right)$$

For each $t$, we can solve the minimization with respect to $\boldsymbol{p} \in \mathbb{R}^n$. The optimal $\boldsymbol{p} = \boldsymbol{p}(t)$ is given by

$$p_i(t) = -tu_i + \begin{cases} (g_i + u_i t)_+ & \text{under } \Omega_{\searrow}(y_i) \\ 0 & \text{under } \Omega_{\vee}(y_i) \\ (g_i + u_i t)_- & \text{under } \Omega_{\nearrow}(y_i). \end{cases}$$

Therefore, $\frac{1}{n} \text{dist}(\boldsymbol{g}, \mathcal{C}(\boldsymbol{u}, \boldsymbol{y}))^2$ can be written as

$$\frac{1}{n} \text{dist}(\boldsymbol{g}, \mathcal{C}(\boldsymbol{u}, \boldsymbol{y}))^2 = \frac{1}{n} \inf_{t \in \mathbb{R}} \sum_{i=1}^n (tu_i + g_i)_-^2 I\{\Omega_{\searrow}(y_i)\} + (tu_i + g_i)^2 I\{\Omega_{\vee}(y_i)\} + (tu_i + g_i)_+^2 I\{\Omega_{\nearrow}(y_i)\}$$

$$= \inf_{t \in \mathbb{R}} \varphi_n(t),$$

where

$$\varphi_n(t) := \frac{1}{n} \sum_{i=1}^n (tu_i + g_i)_-^2 I\{\Omega_{\searrow}(y_i)\} + (tu_i + g_i)^2 I\{\Omega_{\vee}(y_i)\} + (tu_i + g_i)_+^2 I\{\Omega_{\nearrow}(y_i)\}.$$

Notice that $\varphi_n(t)$ is a random and convex function, and by the law of large number,

$$\varphi_n(t) \to^p \varphi(t) = \mathbb{E}[(tU + G)_-^2 I\{\Omega_{\searrow}(Y)\}] + \mathbb{E}[(tU + G)^2 I\{\Omega_{\vee}(Y)\}] + \mathbb{E}[(tU + G)_+^2 I\{\Omega_{\nearrow}(Y)\}]$$

for each $t \in \mathbb{R}$. By Lemma A.1, we have

$$\delta_\infty^{-1} = \inf_{t \in \mathbb{R}} \varphi(t),$$

and $\varphi(t)$ is coercive, i.e., $\lim_{|t| \to +\infty} \varphi(t) = +\infty$. Then, $\inf_{t \in \mathbb{R}} \varphi(t)$ can be reduced to $\min_{t \in K} \varphi(t)$ for a compact set $K \subset \mathbb{R}$, and if a convex function converges point-wisely then it converges uniformly over any compact set. This provides $\inf_{\in \mathbb{R}} \varphi_n(t) \to^P \inf_{t \in \mathbb{R}} \varphi(t) = \delta_\infty^{-1}$ and completes the proof. $\square$

## B. Set up for infinite-dimensional optimization problem

**Lemma B.1.** *Suppose that $\ell_Y : \mathbb{R} \to \mathbb{R}$ is a proper lower semicontinuous convex function. Then the map*

$$\mathcal{L} : \mathbb{R} \times \mathcal{H} \to \mathbb{R} \cup \{+\infty\}, \quad (a, v) \mapsto \begin{cases} \mathbb{E}[\ell_Y(aU + v)] & \text{if } \mathbb{E}[|\ell_Y(aU + v)|] < +\infty \\ +\infty & \text{otherwise} \end{cases}$$

*is again a proper lower semicontinuous convex function. Furthermore, for all $(a, v) \in \text{dom}\, \mathcal{L}$, the subderivative at $(a, v)$ is given by*

$$\partial_a \mathcal{L} = \{\mathbb{E}[Uh] : h \in \partial \ell_Y(aU + v)\} \cap \mathbb{R},$$
$$\partial_v \mathcal{L} = \partial \ell_Y(aU + v) \cap \mathcal{H}.$$

*Proof.* Proposition 16.63 in Bauschke & Combettes (2017). $\square$

**Lemma B.2** (Lemma A.1 of Bellec & Koriyama (2023)). *Define $\mathcal{G} : \mathcal{H} \to \mathbb{R}$ as*

$$\mathcal{G} : \mathcal{H} \to \mathbb{R}, \quad v \mapsto \|v\| - \frac{\mathbb{E}[vG]}{\sqrt{1 - \delta^{-1}}}.$$

*Then $\mathcal{G}$ is convex, Lipschitz, and finite valued. Furthermore, $\mathcal{G}$ is Fréchet differentiable at $\mathcal{H} \setminus \{0\}$ in the sense that $\mathcal{G}(v + h) = \mathcal{G}(v) + \mathbb{E}[\nabla \mathcal{G}(v)h] + o(\|h\|)$ for all $\|v\| > 0$, where the gradient $\nabla \mathcal{G}(v)$ is given by*

$$\nabla \mathcal{G}(v) = \frac{v}{\|v\|} - \frac{G}{\sqrt{1 - \delta^{-1}}}.$$

**Lemma B.3** (Existence of Lagrange multiplier)**.** *Assume* $\mathbb{E}[|\ell_Y(G)|] < +\infty$. *Then* $(a_*, v_*) \in \operatorname{dom} \mathcal{L}$ *solves the constrained optimization problem:*

$$\min_{(a,v) \in \mathbb{R} \times \mathcal{H}} \mathcal{L}(a, v) \quad \text{subject to} \quad \mathcal{G}(v) \leq 0$$

*if and only if there exists an associated Lagrange multiplier* $\mu_* \geq 0$ *such that the KKT condition below is satisfied:*

$$-\mu_* \partial \mathcal{G}(v_*) \cap \partial_v \mathcal{L}(a_*, v_*) \neq \emptyset, \quad 0 \in \partial_a \mathcal{L}(a_*, v_*), \quad \mu_* \mathcal{G}(v_*) = 0, \quad \mathcal{G}(v_*) \leq 0. \tag{15}$$

*Now we further assume that*

- $\mathbb{E}[|\inf_u \ell_Y(u)|] < +\infty$.

- $\ell_Y$ *is strictly convex.*

- $\mathbb{P}(\Omega_\searrow) + \mathbb{P}(\Omega_\nearrow) > 0$ *and* $\ell_Y(\cdot)$ *is not constant with probability* 1.

- *There exists a positive constant b and a positive random variable* $D(Y)$ *such that under* $\Omega_\vee$,

$$\ell_Y(u) \geq b^{-1}|u| - D(Y), \quad \forall u \in \mathbb{R}$$

  *with* $\mathbb{E}[D(Y)^2 I\{\Omega_\vee\}] < +\infty$.

*Then, the Lagrange multiplier is always strictly positive* $\mu_* > 0$ *and the constraint is binding, i.e.,* $\mathcal{G}(v_*) = 0$.

*Proof.* First we verify Slater's condition:

$$\operatorname{lev}_{\leq 0} \mathcal{G} \subseteq \operatorname{int} \operatorname{dom} \mathcal{G}, \quad \operatorname{dom} \mathcal{L} \cap \operatorname{lev}_{<0} \mathcal{G} \neq \emptyset.$$

Since $\mathcal{G}(v) = \|v\| - \mathbb{E}[vG]/\sqrt{1 - \delta^{-1}}$ is finite valued the first condition $\operatorname{lev}_{\leq 0} \mathcal{G} \subseteq \operatorname{int} \operatorname{dom} \mathcal{G}$ immediately follows. As for the second condition, $(a, v) = (0, G)$ satisfies $\mathcal{G}(G) = 1 - (1 - \delta^{-1})^{-1/2} < 0$ and $|\mathcal{L}(a, v)| = |\mathbb{E}[\ell_Y(G)]| \leq \mathbb{E}[|\ell_Y(G)|] < +\infty$ by the assumption. Therefore, the objective function and the constraint $(\mathcal{L}, \mathcal{G})$ satisfy Slater's condition. With Slater' condition, the "if and only if " part follows from Proposition 27.21 of Bauschke & Combettes (2017).

Let us show $\mu_* > 0$. Suppose $\mu_* = 0$. Then, $(a_*, v_*)$ solves $\min_{(a,v)} \mathbb{E}[\ell_Y(aU + v)]$. Now, for any $n \in \mathbb{N}$, define $v_n \in \mathcal{H}$ as

$$v_n = \begin{cases} u_{\min} I\{\ell_Y(u_{\min}) \leq n\} & \text{under } \Omega_\vee \\ -n & \text{under } \Omega_\nearrow \\ n & \text{under } \Omega_\searrow \end{cases} \quad \text{where } u_{\min} \in \operatorname{argmin}_{u \in \mathbb{R}} \ell_Y(u)$$

Note that $v_n$ is in $\mathcal{H}$ since the coercivity assumption implies that under the event $\Omega_\vee$,

$$|u_{\min}| I\{\ell_Y(u_{\min}) \leq n\} \leq b(n + D(Y)) I\{\ell_Y(u_{\min}) \leq n\}$$

and the RHS has a finite second moment under $\Omega_\vee$ by the moment assumption on $D(Y)$. Evaluating the objective function $\mathcal{L}$ at $(a, v) = (0, v_n)$, by the optimality of $(a_*, v_*)$, we are left with

$$\begin{aligned} \mathbb{E}[\ell_Y(a_* U + v_*)] &\leq \mathbb{E}[\ell_Y(0 \cdot U + v_n)] \\ &= \mathbb{E}[\ell_Y(u_{\min}) I\{\ell_Y(u_{\min}) \leq n\} I\{\Omega_\vee\}] \\ &\quad + \mathbb{E}[\ell_Y(0) I\{\ell_Y(u_{\min}) > n\} I\{\Omega_\vee\}] \\ &\quad + \mathbb{E}[\ell_Y(-n) I\{\Omega_\nearrow\}] \\ &\quad + \mathbb{E}[\ell_Y(n) I\{\Omega_\searrow\}]. \end{aligned}$$

Note that each integrand on RHS is uniformly bounded by $|\inf_u \ell_Y(u)| + |\ell_Y(0)|$, where $|\inf_u \ell_Y(u)|$ and $|\ell_Y(0)|$ have finite moments by the assumption and Lemma B.4. Thus, by the dominated convergence theorem, taking the limit $n \to +\infty$, we obtain

$$\begin{aligned} \mathbb{E}[\ell_Y(a_* U + v_*)] &\leq \mathbb{E}[\min_u \ell_Y(u) I\{\Omega_\vee\}] + 0 + \mathbb{E}[\inf_u \ell_Y(u) I\{\Omega_\nearrow\}] + \mathbb{E}[\inf_u \ell_Y(u) I\{\Omega_\searrow\}] \\ &= \mathbb{E}[\inf_u \ell_Y(u)] \end{aligned}$$

and $\mathbb{E}[\ell_Y(a_*U + v_*) - \inf_u \ell_Y(u)] \leq 0$. Since the integrand $\ell_Y(a_*U + v_*) - \inf_u \ell_Y(u)$ is non-negative, we get

$$\ell_Y(a_*U + v_*) = \inf_u \ell_Y(u)$$

with probability 1. Let us consider the event $\Omega_\nearrow$. By the strict convexity of $\ell_Y(\cdot)$, under the event $\Omega_\nearrow$, we have always $\ell_Y(x) > \lim_{t \to +\infty} \ell_Y(-t) = \inf_u \ell_Y(u)$ for all $x$. This means that $\ell_Y(a_*U + v_*) = \inf_u \ell_Y(u)$ cannot occur under $\Omega_\nearrow$, and hence $\mathbb{P}(\Omega_\nearrow) = 0$. By the same argument, we get $\mathbb{P}(\Omega_\searrow) = 0$. This is a contradiction with $\mathbb{P}(\Omega_\nearrow) + \mathbb{P}(\Omega_\searrow) > 0$, so we must have $\mu_* > 0$.

$\square$

**Lemma B.4.** *Suppose $\mathbb{E}[\ell_Y(G)_+] < +\infty$ and $\mathbb{E}[\inf_u \ell_Y(u)_-] > -\infty$ where $G \sim N(0,1)$ independent of $Y$. Then, $\mathbb{E}[|\ell_Y(0)|] < +\infty$.*

*Proof.* Note

$$|\ell_Y(0)| \leq \max(-(\inf_u \ell_Y(u))_-, \ell_Y(0)_+) \leq -(\inf_u \ell_Y(u))_- + \ell_Y(0)_+$$

and the RHS has a finite moment by the assumption and Jensen's inequality $\mathbb{E}[\ell_Y(0)_+] = \mathbb{E}[\ell_Y(\mathbb{E}[G])_+] \leq \mathbb{E}[\ell_Y(G)_+] < +\infty$. Here we have used the fact that $u \mapsto (\ell_Y(u))_+$ is convex and $G \sim N(0,1)$ is independent of $Y$. $\square$

## C. Equivalence between nonlinear system and infinite-dimensional optimization problem: Theorem 3.1

**Lemma C.1.** *Suppose $\mathbb{E}[|\inf_u \ell_Y(u)|] < +\infty$ and $\mathbb{E}[|\ell_Y(G)|] < +\infty$. Then for any $a, \sigma, \gamma \in \mathbb{R} \times \mathbb{R}_{>0} \times \mathbb{R}_{>0}$, $\mathrm{prox}[\gamma \ell_Y](aU + \sigma G) \in \mathcal{H}$.*

*Proof.* Denote $\mathrm{prox}[\gamma \ell_Y](aU + \sigma G)$ by $p_*$. Since $p_*$ minimizes $u \mapsto \frac{1}{2\gamma}(aU + \sigma G - u)^2 + \ell_Y(u)$, we have the upper estimate

$$\frac{1}{2\gamma}(aU + \sigma G - p_*)^2 + \ell_Y(p_*) \leq \frac{1}{2\gamma}(aU + \sigma G)^2 + \ell_Y(0).$$

With $\ell_Y(p) \geq \inf_u \ell_Y(u)$, this implies

$$(aU + \sigma G - p_*)^2 \leq (aU + \sigma G)^2 + 2\gamma(\ell_Y(0) - \inf_u \ell_Y(u)),$$

where the RHS has a finite moment. This means $aU + \sigma G - p_* \in \mathcal{H}$ and $p_* \in \mathcal{H}$. $\square$

**Lemma C.2.** *Suppose $(a_*, v_*) \in \mathbb{R} \times \mathcal{H}$ solves the optimization problem with $\|v_*\| > 0$. Let us take a Lagrange multiplier $\mu_* > 0$ satisfying the KKT condition (15). Define the positive scalar $(\gamma_*, \sigma_*) \in \mathbb{R}^2_{>0}$ by*

$$\sigma_* = \frac{\|v_*\|}{\sqrt{1 - \delta^{-1}}}, \quad \gamma_* = \frac{\sigma_* \sqrt{1 - \delta^{-1}}}{\mu_*}.$$

*Then $v_*$ takes the form of*

$$v_* = \mathrm{prox}[\gamma_* \ell_Y](a_*U + \sigma_*G) - a_*U$$

*and $(a_*, \sigma_*, \gamma_*)$ solves the nonlinear system of equations:*

$$\gamma^{-2}\delta^{-1}\sigma^2 = \mathbb{E}[\ell_Y'(\mathrm{prox}[\gamma \ell_Y](aU + \sigma G))^2] \tag{16}$$

$$0 = \mathbb{E}[U\ell_Y'(\mathrm{prox}[\gamma \ell_Y](aU + \sigma G))] \tag{17}$$

$$\sigma(1 - \delta^{-1}) = \mathbb{E}[G\,\mathrm{prox}[\gamma \ell_Y](aU + \sigma G)] \tag{18}$$

*Proof.* The map $v \mapsto \mathcal{G}(v)$ is Fréchet differentiable at $v = v_* \neq 0$ with $\nabla \mathcal{G}(v_*) = \frac{v_*}{\|v_*\|} - \frac{G}{\sqrt{1 - \delta^{-1}}}$ (Lemma B.2), while the constraint is binding $\mathcal{G}(v_*) = 0$ (Lemma B.3). Thus, we have

$$-\mu_* \nabla \mathcal{G}(v_*) = -\mu_*\left(\frac{v_*}{\|v_*\|} - \frac{G}{\sqrt{1 - \delta^{-1}}}\right) \in \partial_v \mathcal{L}(a_*, v_*), \quad 0 \in \partial_a \mathcal{L}(a_*, v_*), \quad \mathcal{G}(v_*) = 0,$$

By the definition of $(\sigma_*, \gamma_*)$, the condition $-\mu_* \nabla \mathcal{G}(v_*) \in \partial_v \mathcal{L}(a_*, v_*)$ yields

$$\partial_v \mathcal{L}(a_*, v_*) \ni -(v_* - \sigma_* G)/\gamma_*$$

This means that $v_*$ also minimizes the map

$$\mathcal{H} \ni v \mapsto \mathcal{L}(v) + \mathbb{E}\Big[\frac{(\sigma_* G - v)^2}{2\gamma_*}\Big] = \mathbb{E}\Big[\ell_Y(a_* U + v) + \frac{(\sigma_* G - v)^2}{2\gamma_*}\Big].$$

Since $\mathrm{prox}[\gamma_* \ell_Y](a_* U + \sigma_* G) - a_* U$ minimizes the integrand and belongs to $\mathcal{H}$ by Lemma C.1, we have

$$v_* = \mathrm{prox}[\gamma_* \ell_Y](a_* U + \sigma_* G) - a_* U \in \mathcal{H}.$$

With $\sigma G - v_* = \gamma \ell_Y'(v_* + a_* U)$, this also gives $\ell_Y'(v_* + a_* U) \in \mathcal{H}$. This in particular means $\mathbb{E}[U \ell_Y'(a_* U + v_*)] \in \mathbb{R}$. With $\partial_a \mathcal{L}(a, v) = \{\mathbb{E}[U \ell_Y'(aU + v)]\} \cap \mathbb{R}$, the condition $0 \in \partial_a \mathcal{L}(a_*, v_*)$ provides

$$0 = \mathbb{E}[U \ell_Y'(a_* U + v_*)].$$

With $v_* = \mathrm{prox}[\gamma_* \ell_Y](a_* U + \sigma_* G) - a_* U$, we get (17). As for (16) and (18), rearranging $\sigma_* = \|v_*\|/\sqrt{1 - \delta^{-1}}$ and $\mathcal{G}(v_*) = \|v_*\| - \mathbb{E}[v_* G]/\sqrt{1 - \delta^{-1}} = 0$ yields

$$\begin{aligned}
\|\sigma_* G - v_*\|^2 &= \sigma_*^2 - 2\sigma_* \mathbb{E}[v_* G] + \|v_*\|^2 \\
&= \sigma_*^2 - 2\sigma_*^2(1 - \delta^{-1}) + \sigma_*^2(1 - \delta^{-1}) \\
&= \delta^{-1}\sigma_*^2,
\end{aligned}$$
$$\mathbb{E}[G(v_* + a_* U)] = \sigma_*(1 - \delta^{-1}) + 0.$$

Substituting $v_* = \mathrm{prox}[\gamma_* \ell_Y](a_* U + \sigma_* G) - a_* U$ to these two equations, we obtain (16) and (18). This completes the proof. $\qquad\square$

**Lemma C.3.** *Suppose $(a_*, \sigma_*, \gamma_*) \in \mathbb{R} \times \mathbb{R}_{>0}^2$ solves the nonlinear system (16)-(18). Then, $(a_*, v_*) \in \mathbb{R} \times \mathcal{H}$ with*

$$v_* = \mathrm{prox}[\gamma_* \ell_Y](a_* U + \sigma_* G) - a_* U \in \mathcal{H}$$

*solves the infinite dimensional optimization problem with $\|v_*\| = \sigma_* \sqrt{1 - \delta^{-1}} > 0$, and the KKT condition (15) is satisfied by the Lagrange multiplier $\mu_* = \sigma_* \sqrt{1 - \delta^{-1}}/\gamma_* > 0$*

*Proof.* We know from Lemma C.1 that $v_* \in \mathcal{H}$ and $\sigma G - v_* = \gamma \ell_Y'(v_* + a_* U) \in \mathcal{H}$. In this case the subderivaitve of $\mathcal{L}$ at $(a_*, v_*)$ is

$$\partial_a \mathcal{L}(a_*, v_*) = \{\mathbb{E}[U \ell_Y'(a_* + v_*)]\}, \quad \partial_v \mathcal{L}(a_*, v_*) = \{\ell_Y'(a_* + v_*)\}.$$

Noting $\mathbb{E}[UG] = 0$, the nonlinear system can be written as

$$\begin{aligned}
\delta^{-1}\sigma_*^2 &= \mathbb{E}[(\sigma G - v_*)^2] \\
0 &= \mathbb{E}[U \ell_Y'(a_* U + v_*)] \\
\sigma_*(1 - \delta^{-1}) &= \mathbb{E}[G v_*].
\end{aligned}$$

Here, the second equation gives

$$0 \in \partial_a \mathcal{L}(a_*, v_*).$$

Rearranging the first and the third equations, we have

$$\|v_*\|^2 = (1 - \delta^{-1})\sigma_*^2, \quad \mathbb{E}[v_* G] = \sigma_*(1 - \delta^{-1}).$$

This implies $\|v_*\| > 0$ and $\mathcal{G}(v_*) = 0$. Since $\|v_*\| > 0$, $\mathcal{G}$ is differentiable at $v_*$. The derivative formula gives

$$-\frac{\|v_*\|}{\gamma_*} \cdot \nabla \mathcal{G}(v_*) = -\frac{\|v_*\|}{\gamma_*}\Big(\frac{v_*}{\|v_*\|} - \frac{G}{\sqrt{1 - \delta^{-1}}}\Big) = \frac{\sigma_* G - v_*}{\gamma_*} = \ell_Y'(a_* U + \sigma_* G) \in \partial_v \mathcal{L}(a_*, v_*)$$

Therefore, $(a_*, v_*)$ satisfies the KKT condition (15) with the Lagrange multiplier $\mu_* = \frac{\|v_*\|}{\gamma} > 0$, and $(a_*, v_*)$ solves the constrained optimization problem. $\qquad\square$

# D. Non-degeneracy and uniqueness

## D.1. Proof of Lemma 3.2

The argument in this proof is inspired by the proof of Lemma 2.6] in (Bellec & Koriyama, 2023). Suppose $v_* = 0$. Let $\mu_* > 0$ be the associated Lagrange multiplier satisfying the KKT condition (15) so that $v_* = 0$ solves the unconstrained optimization problem $\min_{v \in \mathcal{H}} \mathcal{L}(a_*, v) + \mu_* \mathcal{G}(v)$. With $\mathcal{G}(0) = 0$, this gives

$$\mathbb{E}[\ell_Y(a_* U)] \leq \mathbb{E}[\ell_Y(a_* U + v)] + \mu_*\big(\|v\| - \mathbb{E}[vG]/\sqrt{1 - \delta^{-1}}\big)$$

for all $v \in \mathcal{H}$. Multiplying the both sides by $\lambda := \sqrt{1 - \delta^{-1}}/\mu_* > 0$ and denoting $f(\cdot) = \lambda(\ell_Y(a_* U + \cdot) - \ell_Y(a_* U))$, we have

$$0 \leq \mathcal{A}(v) := \mathbb{E}[f(v)] + \mathbb{E}[v^2]^{1/2}\sqrt{1 - \delta^{-1}} - \mathbb{E}[vG] \quad \text{for all } v \in \mathcal{H}. \tag{19}$$

We parametrize $v \in \mathcal{H}$ as $v_t = \text{prox}[tf](tG) \in \mathcal{H}$ for all $t > 0$ and show $\mathcal{A}(v_t) < 0$ for sufficiently small $t > 0$. Note that $t^{-1}(tG - v_t) \in \partial f(v_t)$ implies

$$-f(v_t) = f(0) - f(v_t) \geq t^{-1}(tG - v_t)(0 - v_t) = -Gv_t + t^{-1}v_t^2.$$

Substituting this to (19), noting that $\mathbb{E}[v_t G]$ is cancelled out, we have

$$\mathcal{A}(v_t) \leq \mathbb{E}[Gv_t - t^{-1}v_t^2] + \mathbb{E}[v_t^2]^{1/2}\sqrt{1 - \delta^{-1}} - \mathbb{E}[v_t G] \leq -t \cdot \|t^{-1}v_t\|\big(\|t^{-1}v_t\| - \sqrt{1 - \delta^{-1}}\big). \tag{20}$$

Now we identify the limit of $\|v_t/t\|$ as $t \to 0+$. The Moreau envelope constructed function $t \mapsto \text{env}_f(tG, t) = \frac{1}{2t}(tG - v_t)^2 + f(v_t)$ has the derivative

$$-\frac{1}{2t^2}(tG - v_t)^2 + \frac{1}{t}G(tG - v_t) = \frac{1}{2}\bigg[G^2 - \bigg(\frac{v_t}{t}\bigg)^2\bigg],$$

which is increasing in $t$ because the Moreau envelope $\text{env}_f(x, y)$ is jointly convex in $(x, y) \in \mathbb{R} \times \mathbb{R}_{>0}$ (cf. Lemma D.1 of Thrampoulidis et al. (2018)). This means that $v_t^2/t^2$ is non-increasing in $t$ and has a non-negative limit as $t \to 0+$. By the monotone convergence theorem, we get

$$\lim_{t \to 0+} \|v_t/t\|^2 = \mathbb{E}[\lim_{t \to 0+}(v_t/t)^2] \in [0, +\infty] \tag{21}$$

Let us compute the limit $v_t/t$. First, we claim $v_t \to 0$. By the optimality of $v_t = \text{prox}[tf](tG)$ with $f(\cdot) = \lambda(\ell_Y(a_* U + \cdot) - \ell_Y(a_* U))$, we have

$$\frac{1}{2t}(tG - v_t)^2 + tf(v_t) \leq \frac{(tG)^2}{2t} + f(0) = tG^2 + 0.$$

This gives

$$\frac{1}{2}(tG - v_t)^2 \leq t^2 G^2 - t\inf_x f(x) = t^2 G^2 - t(\inf_u \ell_Y(u) - \ell_Y(a_* U))$$

Since $(G, \inf_u \ell_Y(u), \ell_Y(a_* U))$ are all bounded in $\ell_1$, they are all finite with probability 1. This provides $\lim_{t \to 0+} v_t = 0$. Combined with $G - v_t/t = f'(v_t) = \lambda \ell_Y'(a_* U + v_t)$, since $\ell_Y(\cdot)$ is $C^1$, we get $v_t/t \to G - \lambda \ell_Y'(a_* U)$. Substituting this limit to (21), we obtain

$$\|v_t/t\|^2 \to \mathbb{E}[(G - \lambda \ell_Y'(a_* U))^2] \in [0, +\infty].$$

Recall that (19) and (20) imply

$$0 \leq \mathcal{A}(v_t)/t \leq -\|v_t/t\|(\|v_t/t\| - \sqrt{1 - \delta^{-1}}), \quad \forall t > 0.$$

This excludes the case $\mathbb{E}[(G - \lambda \ell_Y'(a_* U))^2] = +\infty$ otherwise the RHS converges to $-\infty$ as $t \to 0$. Thus, we must have $\mathbb{E}[(G - \lambda \ell_Y'(a_* U))^2] < +\infty$ and $\ell_Y'(a_* U) \in \mathcal{H}$. Expanding the square, since $G$ and $\ell_Y'(a_* U)$ is independent, we get

$$\lim_{t \to +\infty} \|v_t/t\|^2 = 1 + \lambda^2 \mathbb{E}[\ell_Y'(a_* U)^2].$$

Substituting this to the upper bound of $\mathcal{A}(v_t)/t$,

$$\lim_{t \to 0} -\|v_t/t\|(\|v_t/t\| - \sqrt{1 - \delta^{-1}}) = -\sqrt{(1 + \lambda^2 \mathbb{E}[\ell_Y'(a_* U)^2])}(\sqrt{1 + \lambda^2 \mathbb{E}[\ell_Y'(a_* U)^2]} - \sqrt{1 - \delta^{-1}})$$

$$\leq -(1 - \sqrt{1 - \delta^{-1}}) < 0,$$

which implies that there exists a sufficiently small $t'$ such that $\mathcal{A}(v_{t'})/t' < 0$. This is a contradiction with $\mathcal{A}(v_t)/t \geq 0$ for all $t > 0$. Therefore, we must have $v_* \neq 0$.

### D.2. Proof of Lemma 3.3

Suppose that there exists two minimizer $(a, v), (a', v') \in \mathbb{R} \times \mathcal{H}$. Let $(a_t, v_t) = t(a, v) + (1 - t)(a', v')$ for all $t \in [0, 1]$. Then, by the convexity of objective function and constraint, $(a_t, v_t)$ solves the constrained optimization problem. This implies $\mathcal{L}(a_t, v_t) = t\mathcal{L}(a, v) + (1 - t)\mathcal{L}(a', v')$ for all $t \in [0, 1]$. With $\mathcal{L}(a, v) = \mathbb{E}[\ell_Y(aU + v)]$, we must have

$$\mathbb{E}\Big[\ell_Y(t(aU + v) + (1 - t)(a'U + v')) - t\ell_Y(aU + v) - (1 - t)\ell_Y(a'U + v')\Big] = 0.$$

By the convexity of $\ell_Y$,

$$\ell_Y(t(aU + v) + (1 - t)(a'U + v')) = t\ell_Y(aU + v) + (1 - t)\ell_Y(a'U + v')$$

for all $t \in [0, 1]$. By the strict convexity of $\ell_Y$, we must have $aU + v = a'U + v'$. If $a = a'$ holds then we have $v = v'$, completing the proof of uniqueness. We now show that $a = a'$. The condition $aU + v = a'U + v'$ gives

$$v_t = tv + (1 - t)v' = t\{v + U(a - a')\} + (1 - t)v' = v' + tU(a - a'),$$

so that $\mathbb{E}[v_t G] = \mathbb{E}[v'G]$ by independence of $G$ and $U$. Meanwhile, since $(a_t, v_t)$ solves the constrained optimization problem for all $t \in [0, 1]$, the constraint is satisfied in equality $\|v_t\| - \mathbb{E}[v_t G]/\sqrt{1 - \delta^{-1}} = 0$ for all $t \in [0, 1]$ (see Lemma B.3). Then, $t \mapsto \|v_t\|$ must be constant as well. The polynomial $\|v_t\|^2$ is given

$$\|v_t\|^2 = \|v' + tU(a - a')\|^2 = \|v'\|^2 + t\,\mathbb{E}[v'U(a - a')] + t^2(a - a')^2.$$

Since it is constant in $t$, the quadratic term must be 0, hence $(a - a')^2 = 0$.

Finally, let us show the uniqueness of Lagrange multiplier. Suppose that there exists two distinct Lagrange multipliers $\mu_* \neq \mu_{**} \in \mathbb{R}_{>0}$ associated with the minimizer $(a_*, v_*)$. Since the subderivative $\partial_v \mathcal{L}(a_*, v_*)$ is the singleton $\{\ell_Y'(a_*U + v_*)\}$ at the minimizer $(a_*, v_*)$, the KKT conditions

$$-\mu_* \nabla \mathcal{G}(v_*), -\mu_{**} \nabla \mathcal{G}(v_*) \in \partial_v \mathcal{L}(a_*, v_*)$$

lead to

$$-\mu_* \nabla \mathcal{G}(v_*) = -\mu_{**} \nabla \mathcal{G}(v_*) = \ell_Y'(a_*U + v_*).$$

Combined with $\mu_* \neq \mu_{**}$, this gives $\nabla \mathcal{G}(v_*) = 0$ and $\ell_Y'(a_*U + v_*) = 0$. Here $0 = \nabla \mathcal{G}(v_*) = v_*/\|v_*\| - G/\sqrt{1 - \delta^{-1}}$ implies $v_* = \frac{\|v_*\|}{\sqrt{1 - \delta^{-1}}}G$. Letting $\sigma_* = \|v_*\|/\sqrt{1 - \delta^{-1}} > 0$, substituting this to $\ell_Y'(a_*U + v_*) = 0$, we get

$$\ell_Y'(a_*U + \sigma_*G) = 0$$

Since $\ell_Y$ is strictly convex, this means $\mathbb{P}(\Omega_\vee) = 1$, which is a contradiction with $\mathbb{P}(\Omega_\vee) < 1$. This completes the proof of uniqueness of Lagrange multiplier.

## E. Proof of the phase transition: Theorem 3.4

**Lemma E.1.** *If $\delta \leq \delta_\infty$, the optimization problem* (5) *does not admit any minimizer.*

*Proof.* Let us take $(t_*, p_*) \in \mathcal{C} \subset \mathbb{R} \times \mathcal{H}$ as in Lemma A.1 so that

$$\|p_*\| = \sqrt{1 - \delta_\infty^{-1}}, \quad \mathbb{E}[p_*^2] = \mathbb{E}[p_*G].$$

Substituting this to $\mathcal{G}(p_*) = \|p_*\| - \mathbb{E}[p_*G]/\sqrt{1 - \delta^{-1}}$, using the condition $\delta \leq \delta_\infty$, we have

$$\mathcal{G}(p_*) = \|p_*\|\Big(1 - \frac{\sqrt{1 - \delta_\infty^{-1}}}{\sqrt{1 - \delta^{-1}}}\Big) \leq 0,$$

i.e., $p_*$ satisfies the constraint $\mathcal{G} \leq 0$ under the assumption $\delta \leq \delta_\infty$.

Let us fix $(a, v) \in \text{dom}\,\mathcal{L} \cap \text{lev}_{\leq 0}\,\mathcal{G}$ so that $\mathbb{E}[|\ell_Y(aU + v)|] < +\infty$ and $\mathcal{G}(v) \leq 0$. For any $\nu \geq 1$, consider the convex combination with coefficients $(1 - 1/\nu)$ and $1/\nu$ given by

$$(a_\nu, v_\nu) := (a, v)(1 - 1/\nu) + (1/\nu)(\nu t_*, \nu p_*).$$

Note that $(a_\nu, v_\nu) \to (a, v) + (t_*, p_*)$ almost surely as $\nu \to +\infty$. By the convexity of $\mathcal{G}$, the convex combination $(a_\nu, v_\nu)$ also satisfies the constraint $\mathcal{G}(v_\nu) \leq 0$. On the other hand, the convexity of $\ell_Y$ implies

$$\ell_Y(a_\nu U + v_\nu) \leq (1 - 1/\nu)\ell_Y(aU + v) + (1/\nu) \cdot \ell_Y(\nu t_* U + \nu p_*).$$

Here, by the definition of $v_*$,

$$\ell_Y(\nu t_* U + \nu p_*) = \begin{cases} \ell_Y(\nu(G + Ut_*)_-) & \text{under } \Omega_\nearrow \\ \ell_Y(\nu(G + Ut_*)_+) & \text{under } \Omega_\searrow \\ \ell_Y(0) & \text{under } \Omega_\vee \end{cases}$$

and hence $\ell_Y(\nu t_* U + \nu t_*)$ is uniformly upper bounded by $\ell_Y(0)$ for all $\nu \geq 1$. We have

$$\ell_Y(a_\nu U + v_\nu) \leq (1 - 1/\nu)\ell_Y(aU + v) + (1/\nu)\ell_Y(0). \tag{22}$$

Taking expectation, we get

$$\mathcal{L}(a_\nu, v_\nu) \leq (1 - 1/\nu)\mathcal{L}(a, v) + (1/\nu)\,\mathbb{E}[\ell_Y(0)].$$

Now we consider the limit $\nu \to +\infty$. The RHS converges to $\mathcal{L}(a, v)$ as $\nu \to +\infty$ since we took $(a, \nu) \in \text{dom}\,\mathcal{L}$ and $\mathbb{E}[|\ell_Y(0)|] < +\infty$ by Lemma B.4. As for the LHS $\mathcal{L}(a_\nu, v_\nu) = \mathbb{E}[\ell_Y(a_\nu U + v_\nu)]$, from (22), the integrand is uniformly bounded as

$$|\ell_Y(a_\nu U + v_\nu)| \leq |\inf_u \ell_Y(u)| + |\ell_Y(aU + v)| + |\ell_Y(0)|$$

where the RHS has a finite expectation. Therefore, by the dominated convergence theorem, we get

$$\begin{aligned} \lim_{\nu \to +\infty} \mathcal{L}(a_\nu, v_\nu) &= \lim_{\nu \to +\infty} \mathbb{E}[\ell_Y(a_\nu U + v_\nu)] \\ &= \mathbb{E}\big[\lim_{\nu \to +\infty} \ell_Y(a_\nu U + v_\nu)\big] \\ &= \mathbb{E}[\ell_Y((a + t_*)U + v + p_*)] \qquad \text{by } (a_\nu, v_\nu) \to (a + t_*, v + p_*) \text{ and the continuity of } \ell_Y(\cdot) \\ &= \mathcal{L}(a + t_*, v + p_*). \end{aligned}$$

We have proved that for any $(a, v) \in \text{dom}\,\mathcal{L}$, the inequality $\mathcal{L}(a + t_*, v + p_*) \leq \mathcal{L}(a, v)$ holds.

Now we suppose that a minimizer $(a_{\min}, v_{\min})$ exists. Then it must be unique by Lemma 3.3. Applying the inequality $\mathcal{L}(a + t_*, v + p_*) \leq \mathcal{L}(a, v)$ we have established with $(a, v) = (a_{\min}, v_{\min})$, we must have $t_* = p_* = 0$. Substituting this to the definition of $p_*$, we are left with

$$(G)_- I\{\Omega_\nearrow(Y)\} + (G)_+ I\{\Omega_\searrow(Y)\} = 0.$$

Taking the expectation of this, since $G$ and $Y$ are independent and $\mathbb{E}[(G)_+] = \mathbb{E}[(G)_-] = \sqrt{2/\pi} > 0$, we obtain $\mathbb{P}(\Omega_\nearrow(Y)) + \mathbb{P}(\Omega_\searrow) = 0$, which contradicts the assumption $\mathbb{P}(\Omega_\vee) = 1 - \mathbb{P}(\Omega_\nearrow(Y)) - \mathbb{P}(\Omega_\searrow) < 1$. Therefore, the minimizer does not exist. $\qquad\square$

**Lemma E.2.** *Assume that there exists a positive constant $b$ and positive random variable $D(Y)$ with $\mathbb{E}[D(Y)] < +\infty$ such that*

$$\ell_Y(u) \geq -D(Y) + \frac{1}{b} \times \begin{cases} u & \text{under } \Omega_\nearrow \\ -u & \text{under } \Omega_\searrow \\ |u| & \text{under } \Omega_\vee \end{cases}$$

*Suppose $\delta > \delta_\infty$. Then for any deterministic $\xi > 0$, if $(a, v)$ satisfies $\mathcal{G}(v) \leq 0$ and $\mathcal{L}(a, v) \leq \xi$, then $a + \|v\| \leq C(\xi)$ for some constant depending on $\xi$. Consequently, the objective function of the minimization problem is coercive and admits a minimizer $(a_*, v_*)$ by Proposition 11.15 in (Bauschke & Combettes, 2017).*

*Proof.* By Lemma A.1, $\delta_\infty$ can be represented as $\delta_\infty^{-1} = \min_{(t,p)\in\mathcal{C}} \mathbb{E}[(G - p)^2]$, where $\mathcal{C} \subset \mathbb{R} \times \mathcal{H}$ is the cone defined as

$$(a, v) \in \mathcal{C} \Leftrightarrow (v + aU) \begin{cases} \leq 0 & \text{under } \Omega_\nearrow \\ \geq 0 & \text{under } \Omega_\searrow \\ = 0 & \text{under } \Omega_\vee \end{cases}$$

and the optimal $(t_*, p_*) \in \text{argmin}_{(t,p)\in\mathcal{C}} \mathbb{E}[(G - p)^2]$ satisfies

$$\|p_*\| = \sqrt{1 - \delta_\infty^{-1}} < \sqrt{1 - \delta^{-1}}, \quad \mathbb{E}[p_*^2] = \mathbb{E}[p_* G] \tag{23}$$

where we have used $\delta > \delta_\infty$. Now for all $(a, v) \in \mathbb{R} \times \mathcal{H}$, define $\tilde{v} \in \mathcal{H}$ as

$$\tilde{v} = -aU + \begin{cases} (aU + v)_- & \text{under } \Omega_\nearrow \\ (aU + v)_+ & \text{under } \Omega_\searrow \\ 0 & \text{under } \Omega_\vee \end{cases}$$

so that $(a, \tilde{v}) \in \mathcal{C}$ for all $(a, v) \in \mathbb{R} \times \mathcal{H}$. Furthermore,

$$v - \tilde{v} = \begin{cases} (v + aU)_+ & \text{under } \Omega_\nearrow \\ (v + aU)_- & \text{under } \Omega_\searrow \\ v + aU & \text{under } \Omega_\vee \end{cases} \quad \text{and consequently} \quad \|v - \tilde{v}\| \leq \|v\| + |a|$$

By the condition $\mathcal{L}(a, v) = \mathbb{E}[\ell_Y(aU + v)] \leq \xi$ and the using the variable $D(Y)$ in Assumption 2.5, we know

$$|v - \tilde{v}| \leq D(Y) + b \begin{cases} \ell_Y((v + aU)_+) & \text{under } \Omega_\nearrow \\ \ell_Y((v + aU)_-) & \text{under } \Omega_\searrow \\ \ell_Y(v + aU) & \text{under } \Omega_\vee \end{cases}$$

$$= D(Y) + b\ell_Y(aU + v) + b \begin{cases} \ell_Y(0) - \ell_Y(v + aU) & \text{if } \Omega_\nearrow \text{ and } aU + v \leq 0 \\ \ell_Y(0) - \ell_Y(v + aU) & \text{if } \Omega_\searrow \text{ and } aU + v \geq 0 \\ 0 & \text{otherwise} \end{cases}$$

$$\leq D(Y) + b\ell_Y(aU + v) + b|\ell_Y(0) - \inf_u \ell_Y(u)|$$

so that

$$\mathbb{E}[|v - \tilde{v}|] \leq b\xi + \mathbb{E}[D(Y)] + b(\mathbb{E}[|\ell_Y(0)|] + \mathbb{E}[|\inf_u \ell_Y(u)|]) = C^{(1)}(\xi).$$

By Assumption 2.4, we can take a sufficiently small $u_0 > 0$ such that

$$\Omega_+ := \left\{\{U > u_0\} \cap \Omega_\nearrow\right\} \cup \left\{\{U < -u_0\} \cap \Omega_\searrow\right\} \cup \left\{\{|U| > u_0\} \cap \Omega_\vee\right\}$$

$$\Omega_- := \left\{\{U < -u_0\} \cap \Omega_\nearrow\right\} \cup \left\{\{U > u_0\} \cap \Omega_\searrow\right\} \cup \left\{\{|U| > u_0\} \cap \Omega_\vee\right\}$$

have positive probabilities. Let $p_0 = \min(\mathbb{P}(\Omega_-), \mathbb{P}(\Omega_+)) > 0$.

Suppose $a > 0$. Under $\{U > u_0\} \cap \Omega_\nearrow$, it holds that $aU + v \leq b\ell_Y(aU + v) + D(Y)$ and hence

$$au_0 \leq aU \leq b\ell_Y(aU + v) + D(Y) - v.$$

Under $\{U < -u_0\} \cap \Omega_\searrow$, we have $-(aU + v) \leq b\ell_Y(aU + v) + D(Y)$ and hence

$$au_0 \leq a(-U) \leq b\ell_Y(aU + v) + D(Y) + v$$

Finally, under $\{|U| > u_0\} \cap \Omega_\vee$, we have $|aU + v| \leq b\ell_Y(aU + v) + D(Y) + v$ so that

$$au_0 \leq a|U| \leq |aU + v| + |v| \leq b\ell_Y(aU + v) + D(Y) + |v|.$$

Combining them all together,

$$
\begin{aligned}
au_0 \, \mathbb{P}(\Omega_+) &\le \mathbb{E}[I\{\Omega_+\}au_0] \\
&\le b\,\mathbb{E}[(\ell_Y(aU+v) + D(Y) + |v|)I\{\Omega_+\}] \\
&= b\,\mathbb{E}[(\underbrace{\ell_Y(aU+v) - \inf_u \ell_Y(u)}_{\ge 0} + \inf_u \ell_Y(u) + \underbrace{D(Y) + |v|}_{\ge 0})I\{\Omega_+\}] \\
&\le b\,\mathbb{E}[\ell_Y(aU+v) - \inf_u \ell_Y(u)] + \mathbb{E}[\inf_u \ell_Y(u)I\{\Omega_+\}] + \mathbb{E}[D(Y)] + \mathbb{E}[|v|] \\
&= b\,\mathbb{E}[\ell_Y(aU+v)] - \mathbb{E}[\inf_u \ell_Y(u)I\{\Omega_+^c\}] + \mathbb{E}[D(Y)] + \mathbb{E}[|v|] \\
&\le b\xi + \mathbb{E}[|\inf_u \ell_Y(u)|] + \mathbb{E}[D(Y)] + \|v\|_2.
\end{aligned}
$$

By the same argument, if $a < 0$, considering the event $\Omega_-$, we have

$$
(-a)u_0 \, \mathbb{P}(\Omega_-) \le b\xi + \mathbb{E}[|\inf_u \ell_Y(u)|] + \mathbb{E}[D(Y)] + \|v\|_2.
$$

Combined with $\min(\mathbb{P}(\Omega_+), \mathbb{P}(\Omega_-)) = p_0 > 0$, we have that

$$
|a| \le (u_0 p_0)^{-1}\Big(\xi + \mathbb{E}[|\inf_u \ell_Y(u)|] + \mathbb{E}[D] + \|v\|_2\Big) \le C_1(\xi)(1 + \|v\|_2)
$$

With $\|v - \tilde v\| \le |a| + \|v\|$, we obtain

$$
\|v - \tilde v\| \le C^{(2)}(\xi)(1 + \|v\|_2). \tag{24}
$$

Now we claim that the Pythagorean inequality

$$
\mathbb{E}[(G - p_*)^2] + \mathbb{E}[(p_* - \tilde v)^2] \le \mathbb{E}[(G - \tilde v)^2] \tag{25}
$$

holds. A proof of (25) is as follows. Define for $s \in [0,1]$ the convex combination $v_s = p_*(1 - s) + s\tilde v$ and consider the function

$$
\varphi(s) = \mathbb{E}[(G - v_s)^2] - \mathbb{E}[(G - p_*)^2] - \mathbb{E}[(v_s - p_*)^2].
$$

By the definition of $v_s$, it follows that $\varphi(0) = 0$. Furthermore, $\varphi(s)$ is linear in $s$, as the quadratic term $\mathbb{E}[v_s^2]$ cancels out. On the other hand, the optimality of $(t_*, p_*) \in \operatorname{argmin}_{(t,p)\in\mathcal{C}} \mathbb{E}[(G - p)^2]$ implies that $\mathbb{E}[(G - p_*)^2] \le \mathbb{E}[(G - v_s)^2]$. Combining this with the definition of $\varphi(s)$, we have $\varphi(s) \ge -\mathbb{E}[(v_s - p_*)^2] = -s^2\,\mathbb{E}[(p_* - \tilde v)^2]$. Since $\varphi(s)$ is linear in $s$ and satisfies $\varphi(s) \ge -O(s^2)$ as $s \to 0$, the slope $\varphi'(0)$ must be non-negative. This implies $0 = \varphi(0) \le \varphi(1)$, which is exactly inequality (25).

In both sides of (25), $\mathbb{E}[\tilde v^2]$ cancel out, $\mathbb{E}[G^2] = 1$ and $\mathbb{E}[(G - p_*)^2] = 1 - \|p_*\|^2$ by (23), so that (25) can be rewritten as

$$
-2\,\mathbb{E}[p_* \tilde v] \le -2\,\mathbb{E}[\tilde v G], \qquad \text{i.e.,} \qquad \mathbb{E}[\tilde v(p_* - G)] \ge 0. \tag{26}
$$

From this, for all $v$ satisfying the constraint $\mathcal{G}(v) = \|v\| - \mathbb{E}[vG]/\sqrt{1 - \delta^{-1}} \le 0$,

$$
\begin{aligned}
\Big(\sqrt{1 - \delta^{-1}} - \|p_*\|\Big)\|v\| &\le \mathbb{E}[vG] - \|p_*\|\|v\| && \text{by } \mathcal{G}(v) = \|v\| - \mathbb{E}[vG]/\sqrt{1 - \delta^{-1}} \le 0 \\
&\le \mathbb{E}[v(G - p_*)] && \text{by the Cauchy-Schwarz inequality } \mathbb{E}[p_* v] \le \|p_*\|\|v\| \\
&\le \mathbb{E}[(\tilde v - v)(p_* - G)] && \text{by } \mathbb{E}[\tilde v(p_* - G)] \ge 0 \text{ in (26).}
\end{aligned} \tag{27}
$$

The parenthesis $\sqrt{1 - \delta^{-1}} - \|p_*\|$ on the left is positive thanks to (23). On the right-hand side, considering the event $I\{|p_* - G| > C\}$ for some constant $C > 0$ to be specified later, the Cauchy-Schwarz inequality gives

$$
\begin{aligned}
\mathbb{E}[(\tilde v - v)(p_* - G)] &\le C\,\mathbb{E}[|\tilde v - v|] + \mathbb{E}[I\{|p_* - G| > C\}|p_* - G||\tilde v - v|] \\
&\le C\,\mathbb{E}[|\tilde v - v|] + \sqrt{\mathbb{E}[I\{p_* - G| \ge C\}|p_* - G|^2]} \cdot \|\tilde v - v\|.
\end{aligned}
$$

Since $p_* - G$ is bounded in L2, we can take a sufficiently large $C^{(3)}(\epsilon)$ for all $\epsilon > 0$ such that $\mathbb{E}[I\{|p_* - G| > C^{(3)}(\epsilon)\}|p_* - G|^2] \leq \epsilon^2$. Thus, we have that for all $\epsilon > 0$,

$$\left(\sqrt{1 - \delta^{-1}} - \|p_*\|\right)\|v\| \leq C^{(3)}(\epsilon)\,\mathbb{E}\,|\tilde{v} - v| + \epsilon\|v - \tilde{v}\|_2. \tag{28}$$

Recall the bound $\|v - \tilde{v}\| \leq C^{(2)}(\xi)(1 + \|v\|)$ in (24), so if we take a sufficiently small $\epsilon = \epsilon_0$ such that $\epsilon_0 C^{(2)}(\xi) \leq (\sqrt{1 - \delta^{-1}} - \|p_*\|)/2$, we obtain

$$\frac{1}{2}\left(\sqrt{1 - \delta^{-1}} - \|p_*\|\right)\|v\| \leq C(\epsilon_0)\,\mathbb{E}\,|\tilde{v} - v| + \epsilon_0 C^{(2)}(\xi) \tag{29}$$

Since $\mathbb{E}\,|\tilde{v} - v| \leq C^{(1)}(\xi)$ was established at the beginning, this completes the proof. $\qquad\square$

