# OpenReview forum: "Phase transitions for the existence of unregularized M-estimators in single index models"
_ICML.cc/2025/Conference — ICML 2025 poster_

### Official Review · Reviewer_tCFd · 2025-02-28

**Overall Recommendation:** 4

**Summary:**

This work considers the problem of the existence of M-estimators in the proportional high-dimensional where the number of samples $n$ and covariate dimensions $p$ diverge at fixed rate $n/d\to\delta$. The main result is to establish a sharp frontier $\delta_{\infty}$ separating regimes where the probability of existence of a minimizer is asymptotically zero or one. This is achieved by exploiting a duality between the existence problem and an optimization problem in a Hilbert space, for which standard techniques apply.

**Claims And Evidence:**

All the claims are followed by rigorous mathematical proofs. Numerical simulations are also provided as an illustration of the result in particular cases.

**Essential References Not Discussed:**

The only missing works I would add are: [1] which was the first work to show that $n=2*d$ Gaussian points can be linearly separated at large $d$ and [2] who generalized this result to linear separators with a given fixed margin. They are not essential, but are foundational works concerning the question addressed here.

- [1] Thomas M Cover. Geometrical and statistical properties of systems of linear inequalities with applica-
tions in pattern recognition. IEEE transactions on electronic computers, (3):326–334, 19651964

- [2]  E Gardner and B Derrida. Optimal storage properties of neural network models. 1988 J. Phys. A: Math. Gen. 21 271

**Experimental Designs Or Analyses:**

N/A.

**Methods And Evaluation Criteria:**

N/A.

**Other Comments Or Suggestions:**

I would suggest the authors add a small discussion on the statistical implications of their findings. For instance, from the formula for $\delta_{\infty}$, can you say anything on the class of link functions which lead to higher or smaller threshold? What phenomenology can we draw out from the theoretical result?

**Other Strengths And Weaknesses:**

Overall, I think this is a solid and well-written contribution. It clearly states the results, how it fits the literature and delivers what is promised, without overclaiming.

Of course, this work has all sorts of the typical limitations related to exact asymptotic works (particular data distribution, convexity of the risk, etc.). But in my opinion these are all minor.

**Questions For Authors:**

Perhaps this is a naif question, but where does the limitation $\delta\in(1,\infty)$ come from in the proof? Is it clear that $\delta_{\infty}>1$ for any convex loss function?

**Relation To Broader Scientific Literature:**

Although there is not a classical "Related Works" section, the authors do a good job in situating their results within the literature. Their work is motivated by (Sur & Candès 2019; 2020), and build on previous contributions from (Montanari et al. 2023; Bellec & Koriyama 2023; Thrampoulidis et al. 2018), which are extensively acknowledged.

**Theoretical Claims:**

I did not carefully check the proofs, but I did skim through them and did not find any red flag. The proof scheme builds on well-known previous work in the literature.

---

> ### Author Rebuttal · Authors · 2025-03-25
>
> Thank you for the additional references.
> We agree that the phase transition phenomena studied in our paper are connected to these earlier works in statistical physics. We addressed this point in our rebuttal to reviewer LDvE; in a way our results are complementary and fill a gap. We will add the references given in the review to the camera-ready version.
>
> **Regarding the assumption $\delta>1$**
>
> The assumptions $\delta>1$ appears in several places. First, we used the assumption $\delta>1$ for the infinite-dimensional optimization problem in equation (5) to be well-defined, where the constraint is given by $\|v\| - \mathbb{E}[vG]/\sqrt{1-\delta^{-1}}\le 0$.
>
> Second, $\delta=1$ allows for instance $p=n$ whence $X\in R^{n\times p}$ is almost surely invertible.
> In this case, for a given $y_i$, one can always find a vector $b\in R^p$ such that $x_i^Tb$ has a given sign for each $x_i$, so the problem is trivially separable.
>
> Finally, $\delta\le 1$ allows for instance $p\ge n+1$, in which case the M-estimator (if it exists) is not unique (i.e., several solutions to the minimization problem in $R^p$ exists). Further structural assumptions are then needed to have a well-defined M-estimator, for instance by considering the solution with minimal L2 norm.
>
> **On question regarding the lower bound $\delta_\infty \ge 1$**:
>
> It is always true for any convex function and response structure of $(x_i, y_i)$ as long as assumption 2.4 is satisfied. This is because the threshold $\delta_\infty$ is defined as
> $$
> \delta_\infty = 1/ \Bigl(\inf_{t\in \mathbb{R}}\varphi(t)\Bigr) \quad \text{with} \quad \varphi(t) = \mathbb{E}[(G+Ut)^2 I_{\Omega_\vee(Y)}]  + \mathbb{E}[(G+tU)^2_+  I_{\Omega_↗(Y)}]  + \mathbb{E}[(G+tU)^2_-  I_{\Omega_↘(Y)}].
> $$
> Here, by $x_{-}^2 \le x^2$ and $x_{+}^2\le x^2$, $\varphi(t)$ can be upper bounded as
> $$
> \varphi(t) \le \mathbb{E}[(G+tU)^2] = 1 + t \mathbb{E}[U^2], \quad \forall t\in \mathbb{R},
> $$
> so taking $\inf_{t\in \mathbb{R}}$ gives the upper estimate $\inf_{t\in \mathbb{R}} \varphi(t) \le 1$. Combined with the definition of $\delta_\infty$, we get $\delta_\infty \ge 1$.
>
> In terms of statistical interpretation, the threshold $\delta_\infty$ decreases when the loss becomes coercive for more realizations of $Y$. This is because $(G + tU)^2$ dominates the other terms $(G + tU)^2_+$ and $(G + tU)^2_-$ pointwise, so $\varphi(t)$ increases with $1_{\Omega_\vee(Y)}$ for each fixed $t$, thereby reducing $\delta_\infty = 1 / \inf_t \varphi(t)$.
>
> This behavior is illustrated in Figure 3. For the Binomial loss with $q$ repeated measurements, $\Omega_\vee(Y)$ is equal to the event $\{0 < Y < q\}$. Consequently, $\delta_\infty$ decreases with larger $q$, as observed in the figure.
>
> We will incorporate the above discussion into the camera-ready version.

---

### Official Review · Reviewer_om5X · 2025-03-04

**Overall Recommendation:** 4

**Summary:**

This paper studies the phase transitions for M-estimators in single index models. Prior work has demonstrated that there exist a threshold $\delta_{\infty}$, such that when $n/p \to \delta$, then the M-estimator exists with high-probability when $\delta > \delta_{\infty}$ while
the M-estimator does not exist when $\delta < \delta_{\infty}$. However the prior results only apply to binary logistic regression, while this work generalize this result to other single-index models.
Another contribution of this paper is that, there is a corresponding nonlinear system which governs the asymptotic behaviour of the M-estimator, but the existence of solution to this system for $\delta > \delta_{\infty}$ is unproven. The authors address this gap by proving the existence of such solution when $\delta > \delta_{\infty}$.

**Claims And Evidence:**

Yes, the two major claims are supported by clear and convincing evidence.
1. the M-estimator exists with high-probability when $\delta > \delta_{\infty}$ while
the M-estimator does not exist when $\delta < \delta_{\infty}$ for general single index models.
The claim is justified by theorem 2.6.

2. the existence of the solution to the critical linear system when $\delta > \delta_{\infty}$.
The claim is justified by theorem 2.7.

**Essential References Not Discussed:**

I do not find essential related works not discussed.

**Experimental Designs Or Analyses:**

The experiments in section4 is sound and supports theorem 2.6. Theorem 2.7 is not supported by experiments.

**Methods And Evaluation Criteria:**

This paper does not involve evaluation.

**Other Comments Or Suggestions:**

Is it possible that in the introduction, you briefly introduce how single index model is applied in machine learning? And why the phase transition is important.

**Other Strengths And Weaknesses:**

Strengths:
The authors generalize the theory of phase transition in single index models from binary logistic regression to general models,  which is of good significance. The theory for the existence of solution regarding the nonlinear systems is also of great importance.

Weaknesses:
no obvious weaknesses.

**Questions For Authors:**

What will happen when $\delta = \delta_{\infty}$? Are there any more detailed characterization of the existence of the M-estimator under this border line scenario?

**Relation To Broader Scientific Literature:**

I am not an expert in this area, so I am not sure how the theory for single index model can affect the broader community.

**Theoretical Claims:**

I have not check the correctness in detail. But the reasoning in the main text, where the authors relate the proof of Theorem 2.7 to infinite dimensional optimization problems seems to be reasonable.

---

> ### Author Rebuttal · Authors · 2025-03-25
>
> Thanks for your suggestion regarding the introduction.
> The single index model is a  flexible yet interpretable framework for modeling nonlinear relationships while avoiding the curse of dimensionality.
> Single models are useful because it is a weak assumption regarding the modeling of $y_i\mid x_i$. It only assumes the existence of a ground truth $\beta^*\in \mathbb R^p$ (or $w$ in our paper), and that the response $y_i$ only depends on $x_i$ through the inner product with the ground truth $\beta^*$. But makes no assumption on how $y_i$ depends on $x_i^T\beta^*$, in particular any nonlinearity is allowed.
>
> For machine learning in proportional asymptotics more specifically, single index models (or more generally multi-index models) allow to characterize the limiting behavior of estimators by a low-dimensional system of equations, for example equation (2) in the submission. Techniques in the submission push the understanding of these systems of equations (sometimes referred to as Gaussian Equivalent Model) that arise from CGMT (Thrampoulidis et al, 2015) or Approximate Message Passing analysis. Without the single-index (or multi-index) model assumption, for instance if $y_i$ depends on $x_i$ through a growing number of dimensions as $n,p\to+\infty$, we are not aware of an established Gaussian Equivalent Model.
>
> **Regading what happens if $p/n\to\delta$ for $\delta=\delta_\infty$**: It is known that phase transitions arising from the convex geometry arguments of Amelunxen et al (2013) have a $1/sqrt n$ width of uncertainty. More concretely, the top of page 5
> https://arxiv.org/abs/1804.09753 (Sur and Candes, 2018) explains this as follows: for any sequence $\lambda_n\to+\infty$,
>
> - if $p/n > \delta_\infty^{-1} + \lambda_n /\sqrt n$ then $P(\text{M-estimator exists})\to 0$,
> - if $p/n < \delta_\infty^{-1} - \lambda_n /\sqrt n$ then $P(\text{M-estimator exists})\to 1$.
>
> So it is possible to be more precise and study what happens near the threshold as long as $p/n$ stays a little more than $1/\sqrt n$ away from the critical treshold. But this phenomenon was known since Amelunxen et al (2013) and  Sur and Candes (2018), so we did not emphasize that in the submission. We will add a remark explaining this, including the above pointers, in the camera-ready version.

---

### Official Review · Reviewer_kj91 · 2025-03-14

**Overall Recommendation:** 4

**Summary:**

This paper investigates the existence of solutions to the nonlinear system of equations that characterize the asymptotic behavior of the M-estimator. Notably, the existence of a solution for $\delta>\delta_{\infty}$ remains largely unproven when the assumption of independence between $x_i$ and $y_i$ is removed in binary logistic regression. The authors extend the existing theory in two key directions: (1) they generalize the results from binary logistic regression to a broader class of single-index models (strictly convex loss), and (2) they analyze the problem beyond the global null case, providing a necessary and sufficient condition that ensures the system admits a solution. This result effectively addresses a gap in the existing literature.

**Claims And Evidence:**

Claims are well supported.

**Essential References Not Discussed:**

n/a

**Experimental Designs Or Analyses:**

The experiments are straightforward and effectively validate the soundness of the results.

**Methods And Evaluation Criteria:**

As this paper generalizes the results of Candès & Sur (2020) from binary logistic regression to a broader class of single-index models, the authors provide numerical validation of their methods by generating data from a Binomial distribution, which can be easily transformed into binary logistic regression by setting $q=1$. Their method was verified and shown to be consistent with the results of Candès & Sur (2020).

**Other Comments Or Suggestions:**

n/a

**Other Strengths And Weaknesses:**

I find this paper to be well-written and clearly articulated in its contributions. To the best of my knowledge, it is original in addressing a theoretical gap in the phase transition literature by extending the theory beyond the assumption of independence between $x_i$ and $y_i$,  particularly beyond the global null setting.

**Questions For Authors:**

n/a

**Relation To Broader Scientific Literature:**

Candès and Sur (2020) primarily examine phase transitions in high-dimensional binary logistic regression models. A key contribution of this paper is its generalization of their results from binary logistic regression to a broader class of single-index models. Additionally, this work advances the theory beyond the global null, addressing a gap left by Sur and Candès (2019), which focused only on numerical solutions.

**Theoretical Claims:**

The main proof strategy was presented in the main text. And I briefly read the proof in the supplementary. The theoretical claims look solid to me though I didn't verify every single piece of the details in the proof.

---

> ### Author Rebuttal · Authors · 2025-03-31
>
> Thanks for the careful reading of the paper and the kind words. We will be happy to provide clarifications if needed in later discussions.

---

### Official Review · Reviewer_LDvE · 2025-03-15

**Overall Recommendation:** 3

**Summary:**

This paper studies phase transitions for the existence of unregularized M-estimators in single-index models under proportional asymptotics, where the sample size n and feature dimension p grow proportionally with n/p → δ ∈ (1, ∞). The authors generalize results previously established for binary logistic regression by Candes & Sur (2020) to more general loss functions and single-index models. They derive an explicit critical threshold δ∞ governing the phase transition for the existence of the M-estimator and rigorously prove that the corresponding nonlinear system admits a solution if and only if δ > δ∞. The work relies on convex geometry, the Gaussian kinematic formula, and infinite-dimensional optimization techniques to establish these results.

**Claims And Evidence:**

These are theorems, with rigorous proofs.

**Essential References Not Discussed:**

The author however seems to omit some of the early foundational studies in information theory and statistical physics, which introduced similar phase transitions long before the modern convex-geometry-based approaches. In particular, the classical results of Cover (1965) on the geometry of linear inequalities and Gardner & Derrida (1988) on the storage capacity of neural networks are directly related to the question of M-estimator feasibility in high dimensions. Moreover, Krauth & Mézard (1987) already presented a phase diagram remarkably similar to the one studied here, predating modern convex-analytic approaches by decades. It would be valuable for the authors to acknowledge these contributions and place their results within this broader historical context.

In more recent year, i can think of few generalization of Surr results, in e.g. "Mignacco, F., Krzakala, F., Lu, Y., Urbani, P., & Zdeborova, L. (2020, November). The role of regularization in classification of high-dimensional noisy gaussian mixture. In International conference on machine learning (pp. 6874-6883). PMLR." which might be covered by the author theorem, or the random feature version of separbality in
Gerace, F., Loureiro, B., Krzakala, F., Mézard, M., & Zdeborová, L. (2020, November). Generalisation error in learning with random features and the hidden manifold model. In International Conference on Machine Learning (pp. 3452-3462). PMLR.

**Experimental Designs Or Analyses:**

N/A

**Methods And Evaluation Criteria:**

N/A

**Other Comments Or Suggestions:**

See above

**Other Strengths And Weaknesses:**

Overall, while the results are rigorous and mathematically insightful, their practical significance for ICML could be debated.After all the main contribution is a proof a known and accepted results. While this is a welcome addition, it is not clear that ICML is the best venue. I nevertheless support publication.

**Questions For Authors:**

See above

**Relation To Broader Scientific Literature:**

The paper does a good job covering prior work in high-dimensional inference, particularly the literature stemming from Candes & Sur (2020) and related works in convex optimization.

**Theoretical Claims:**

These are theorems, with rigorous proofs. The mathematical derivations are elegant and rigorous. The results clarify and extend known phase transition phenomena in M-estimation.

---

> ### Author Rebuttal · Authors · 2025-03-25
>
> We thank the reviewer for highlighting the importance of earlier foundational work from the information theory and statistical physics communities. We agree that the phase transition phenomena studied in our paper are connected in spirit to classical results such as Cover (1965) on the geometry of linear inequalities, which leads to the threshold $\delta_\infty=2$ when the response $y_i$ follows the Bernoulli (1/2) and feature $x_i$ are independent from $y_i$. We also appreciate the pointer to earlier works in statistical physics: Gardner & Derrida (1988), Krauth & Mézard (1987).  We will revise the manuscript to include the historical overview, explicitly acknowledging these contributions.
>
> We also thank the reviewer for drawing our attention to more recent developments, including Mignacco et al. (2020) and Gerace et al. (2020). While our focus is on the existence of unregularized M-estimators for single-index models with Gaussian design, these papers explore related questions and offer complementary perspectives.
>
> The threshold in equation (27) Mignacco et al. (2020) is not directly recovered by our results, since we assume here $x_i\sim N(0, I_p)$ which is not satisfied in the Gaussian mixture setting of Mignacco et al. (2020). However, the techniques presented in the submission can be used to prove that the system in Mignacco et al. (2020) has a solution, and we will add a remark to that effect in the camera-ready version.
>
> Regarding whether machine learning conferences such as ICML are suitable, we believe ICML and similar venues strive to present mathematical results with no gaps. The existence of solutions to the system of equations governing estimators in proportional asymptotics has been typically assumed in the theorems of the corresponding publications, for instance the CGMT paper (Thrampoulidis et al. 2015), Sur and Candes (2018) or Salehi et al. (2019), and several others assume in the theorems that the system of equations of interest admit a unique solution. Existence of solutions to the system can be easily deduced under a strong convexity assumption (e.g., with an additional L2 penalty $\lambda\|\cdot\|_2^2$), but taking the the limit $\lambda\to 0$ as in Appendix B.3 of https://arxiv.org/abs/2201.13383 to deduce results from the strongly convex case $(\lambda>0)$ to the $\lambda=0$ case by continuity requires that the low-dimensional system corresponding to $\lambda=0$ has a unique solution. Thus our method to prove that the system has a unique solution complements these works, and in a sense fills the gap and completes the argument.
>
> Thanks again for the careful reading and pointers to the missing references.

---

### Decision · Program_Chairs · 2025-05-01

**Decision:**

Accept (poster)

**Comment:**

This paper studies phase transitions of existence of M-estimators in single index models in the proportional asymptotic regime. Authors assume isotropic Gaussian input and let the sample size n and feature dimension p to infinity while their ratio converges to a constant \delta. Authors generalize the results of Candes and Sur 2020 to other single index models and find a threshold for \delta that determines the existence of the M-estimator.


This paper was reviewed by four expert reviewers the following Scores: Accept, Accept, Accept, Weak Accept. I think the paper is studying an interesting topic and the results are relevant to ICML community. The following concerns were brought up by the reviewers:

- Some foundational references are missed. These should be cited and discussed in detail.


Authors should carefully go over reviewers' suggestions and address any remaining concerns in their final revision. Based on the reviewers' suggestion, as well as my own assessment of the paper, I recommend including this paper to the ICML 2025 program.